# MR. Video: MapReduce as an Effective Principle for Long Video Understanding

**Ziqi Pang**    **Yu-Xiong Wang**
University of Illinois Urbana-Champaign
{ziqip2, yxw}@illinois.edu

## Abstract

The fundamental challenge of long video understanding, *e.g.*, question answering, lies in the extensive number of frames, making it infeasible to *densely understand the local details* while *comprehensively digest the global contexts*, especially within a limited context length. To address this problem, our insight is to process short video segments individually and combine these segment-level analyses into a final response. This intuition is noted in the well-established MapReduce principle in big data processing and is naturally compatible with inference scaling at the system level. Motivated by this, we propose **MR. Video**[1], a long video understanding framework adopting the MapReduce principle. We define the standard operations of MapReduce in a long video understanding context: the Map steps conduct **independent** and **sequence-parallel** dense perception on short video segments, covering local details, while the Reduce steps **comprehensively aggregate** the segment-level results into an answer with global contexts. Thanks to the low cost and convenience of building video agents, we instantiate such Map and Reduce operations as an effective video agent capable of attending to local details and global contexts. Based on such abilities, we further introduce two critical yet previously under-explored long video understanding designs: (a) *consistent character/object names* in the captions, benefiting the reasoning of actions and stories across long horizons; (b) *question intention analysis*, which changes the key-frame retrieval in previous video agents to localizing the relevant information via jointly reasoning the whole video contexts and questions. Our MR. Video achieves a $>$**7% accuracy improvement** on the challenging LVBench over state-of-the-art video agents and vision-language models (VLMs) and demonstrates a clear advantage on multiple long video benchmarks, highlighting the potential of the MapReduce principle. The code is at https://github.com/ziqipang/MR-Video.

## 1 Introduction

Considering a challenging example for long video understanding (Fig. 1, left): suppose we are watching a fast-paced sports video and wanting to count the number of specific events, *e.g.*, goals by a player, a model should carefully go through every action to inspect the criteria of "a goal by No. 11," and then comprehensively aggregate across the whole video duration, especially when the number of events is as large as 200. Such an example reveals the fundamental challenge in long video understanding: how to *digest global contexts* while *perceiving local details*.

Unfortunately, existing sequence-to-sequence vision-language models (VLMs) [16, 17, 24, 26, 27, 28, 61] that rely on using large language models (LLMs) to process video tokens are limited in context lengths. So they are forced to sample frames sparsely or compress tokens (Fig. 1(a)), losing the dense local details, *e.g.*, missing the events in the example video or failing to recognize the correct person. Although video agents [8, 42, 45, 53] emerge to bypass the VLMs' context length limitations via strategically selecting a small set of video clips to perceive, they sacrifice the other aspects of long video understanding: (1) they generally rely on multi-round exploration of video segment selection

---

[1]pronounced as "mister video"

39th Conference on Neural Information Processing Systems (NeurIPS 2025).

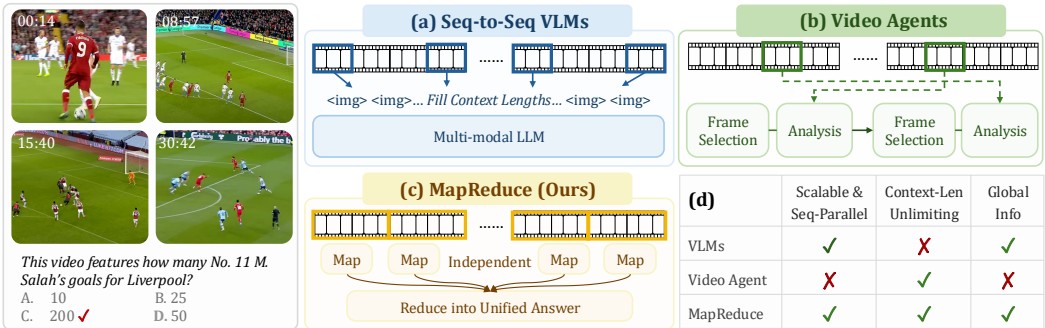

Figure 1: **MapReduce Principle.** Long video understanding requires both *global comprehension* and *detailed perception* without making assumptions about the videos, as shown in the example of counting a large number of events. For such needs, (a) VLMs and (b) video agents are sub-optimal in terms of context lengths, sequential parallelization, and using global context information. (c) We introduce the MapReduce principle and define the Map operation as dense and independent short segment perception, and the Reduce operation as global aggregation, (d) effectively achieving both inference scalability and improved performance.

(Fig. 1(b)), which harms *sequence-parallel* video perception, consequently, inference scalability for extremely long videos; (2) the explicit key segment retrieval contradicts *global comprehension* and might result in insufficient information. Take the example video in Fig. 1, for instance, the existence of 200 events breaks the basic assumption of key segment selection; and even if the model realizes the target of 200 key segments, the iterative search would cost a significant amount of time.

Our key insight to bridge detailed perception and global comprehension lies in scalably decomposing long video understanding into shorter context lengths: the model densely perceives the individual short video clips in parallel, then aggregates the condensed perception results across the whole video (Fig. 1(c)). This framework is noted in how large volumes of data are handled efficiently via the MapReduce principle in distributed systems [7]: we now define the Map step as parallel perception of short clips, and then the Reduce step as global aggregation for the whole video. Such native compatibility with the MapReduce principle also makes our framework friendly for inference scaling at deployment (Fig. 1(d)).

Given the low cost and convenience of building video agents, we instantiate the MapReduce principle via a video agent [8, 42, 45, 53] called **MR. Video**. This also aligns with the trend of utilizing the foundation models in zero-shot to address challenging visual reasoning problems [11, 36]. To unleash the capability of MapReduce, our design of MR. Video introduces two critical yet previously under-explored aspects of long video understanding: (A) The *captioning* stage generates texts as an efficient video analysis medium, where MR. Video specially employs a Reduce step to provide *consistent character/object names* across the long videos, which benefits reasoning across long stories. (B) The *analysis* stage conducts question-related comprehension of the video. MR. Video uniquely advocates using *question intention analysis* to replace the key-segment retrieval in conventional video agents, which does not make any assumptions about the video and provides more comprehensive contexts for complicated multi-hop reasoning.

With both the MapReduce principle and long video agent designs, MR. Video achieves strong performance. Notably, on LVBench [39], one of the most challenging benchmarks featuring hour-long videos and diverse questions, our MR. Video achieves a more than *7% accuracy improvement* over other VLMs and video agents, along with advantages on several other video benchmarks.

To summarize, our contributions are:

1. We introduce the MapReduce principle from the distributed system domain to long video understanding, offering a conceptual framework that mitigates the context length, sequence-parallel scaling, and global context limitations of previous VLMs and video agents.
2. We design "**MR. Video**," a video agent featuring multiple MapReduce stages that generate character-consistent captions and conduct question-intention analysis, both essential for long video reasoning.
3. We highlight the strong performance of MR. Video across multiple long video benchmarks, notably represented by the challenging LVBench. These results suggest the potential of MapReduce as a general principle for long video understanding.

## 2 Related Work

**VLMs for Video Understanding.** Existing VLMs [3, 4, 20, 21, 22, 23, 24, 29, 30, 32, 33, 34, 35, 38, 40, 43, 44, 46, 48, 49, 50, 51, 61, 64] commonly follow LLaVA [28] by projecting image tokens to LLMs. As an image typically takes over 100 tokens in a standard LLaVA model, context lengths become the major challenge for these models in long video understanding: *how to digest the whole video without missing details*? LongVILA [50]'s solution is increasing the context length, but it inherently needs more resources and is still limited by context lengths. Another prevalent solution is decreasing the average tokens per frame via merging or pruning. Such compression can follow certain priors, *e.g.*, similarity of features [4, 22, 32, 34, 35, 40, 46, 48, 49], or Q-former-like [13, 14, 18, 19] learnable module [23]. Notably, the recent VideoChat-Flash [22] can support up to 10k frames with sufficient hardware. However, aggressive compression might lead to unreliable perception of visual details. Such inherent context length limitations of VLMs necessitate more flexible *agentic paradigms* as explained below.

**Video Understanding Agents.** Video agents provide a meta-level LLM controller on the top of VLMs, which splits a long video into sub-tasks of short videos [8, 42, 45, 53, 60]. Therefore, they are not constrained by context lengths. By imitating how humans watch videos, video agents can be treated as increasing the test-time compute of VLMs via multi-round exploration [53], key-frame retrieval [8], and tool-use [42]. However, video agents still demonstrate disadvantages compared with VLMs, as mentioned in Sec. 1: (1) the sequential multi-round exploration hinders scalability, and (2) reliance on key-frame retrieval constrains the understanding of sufficient contexts. From such aspects, MR. Video bridges these gaps with the sequence-parallel Map steps and globally aggregating Reduce steps, respectively (as in Fig. 1(c)).

**LLM Agents.** Our MR. Video, in the context of long video understanding, also contributes to a broader field of research addressing complex problems with the advanced reasoning ability of LLM agents, such as software engineering [15, 52] and knowledge retrieval and reasoning [54, 55, 62]. In addition, our work aligns with the ongoing efforts to explore the zero-shot capabilities of foundation models in various visual reasoning tasks by designing the prompts without training the models, as exemplified by Visual Programming [11], ViperGPT [36], and Socratic Models [57]. With the significant accuracy improvement achieved by our MR. Video, we demonstrate that LLM agents provide an effective way to explore new frameworks at academia-friendly costs.

## 3 Method

### 3.1 Overview

Although the MapReduce principle is widely applicable for handling large volumes of data with scalability, designing the concrete Map and Reduce operations for the specific task of long video understanding is non-trivial. With the convenience and low costs of LLM agents, we create the prompts and workflows of VLMs to address the challenge of *digesting global contexts* while *perceiving local details* in long videos. This leads to an effective video agent: **MR. Video**.

MR. Video's overview[2] is in Fig. 2. It contains two MapReduce stages. (A) The "*Captioning*" stage (Sec . 3.2) generates dense captions, which provide a concise comprehension of the video contents and serve as an efficient medium for answering multiple questions on the same video. (B) The "*Analysis*" stage (Sec. 3.3 and Sec. 3.4) conducts question-specific perception of the video. It first emphasizes understanding the intention of the question (Sec. 3.3), *i.e.*, "what the question is *actually* asking," and then purposefully inspects the visual details or longer temporal spans (Sec. 3.4). The Map steps are independent and sequence-parallel in both stages for different video segments, and the Reduce steps condense the segment-level results into unified video-level understanding.

**Key Operations.** We propose two operations that specially tailor the MapReduce for long video understanding and demonstrate beneficial behaviors unobserved by previous video agents. (1) *Consistent characters/objects in captions*. Instead of purely relying on captioning models, we construct workflows to improve the consistency of character names across a long video, which is beneficial to reasoning across long temporal spans. (2) *Question intention analysis*. We advocate combining the video contexts and questions to understand the goal of the question, such as "*when, why, what*," instead of relying on the key-segment retrieval adopted by previous video agents. By using thorough video contexts, our question intention analysis provides more comprehensive information.

---

[2]The displayed video is the 1st from LVBench (video link). We will consistently use it for method demonstrations for readers' convenience.

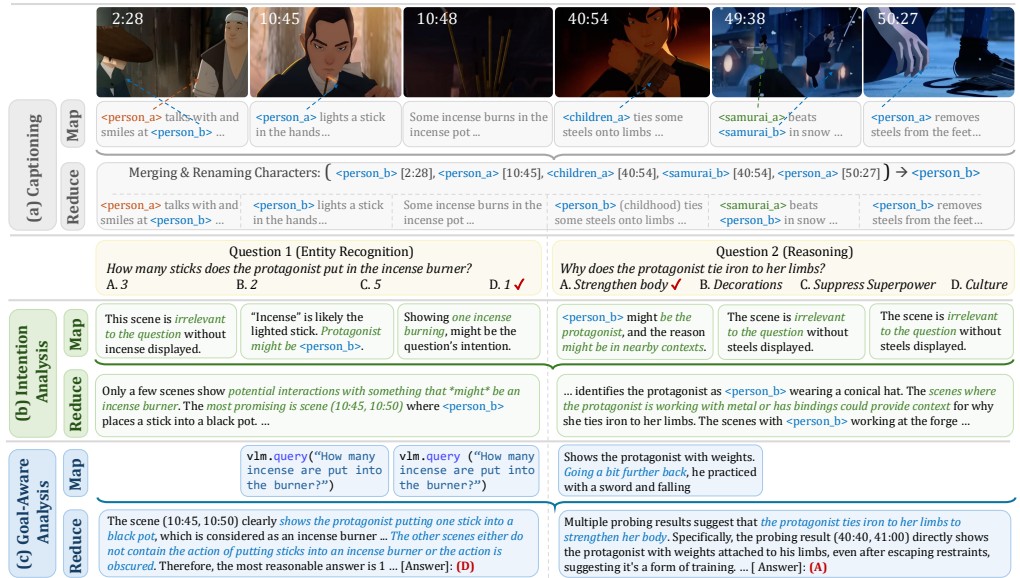

Figure 2: **Overview.** MR. Video reformulates the MapReduce principle into three specialized stages, each designed to address a unique challenge in long video understanding. We show two types of questions focusing on visual details (left) and reasoning (right). **(a)** *Captioning* (Sec. 3.2) generates detailed captions and uniquely enhances the consistency of characters/objects names with the Reduce step, which is repeatedly useful for downstream analysis. **(b)** *Question Intention Analysis* (Sec. 3.3) departs from the conventional key segment retrieval adopted by previous video agents. Instead, it digests the whole question and video content to provide a comprehensive context for detailed perception. **(c)** *Goal-Aware Analysis* (Sec. 3.4) delves deep into detailed perception and reasoning over short and long temporal spans. (For clarity, MR. Video's intermediate texts are simplified.)

## 3.2 Captioning

Captions provide an efficient medium for video understanding that covers long-range contexts. Our captioning is shown in Fig. 2(a): (1) The Map step (Sec. 3.2.1) generates dense captions at the scene level independently, and (2) the Reduce step (Sec. 3.2.2) provides coherent names for repeated characters and objects for consistency. For a 1 – 2 hr video, our captioning generates 500 – 2k captions for the whole video, similar to an article.

Compared with previous video agents [8, 42, 45] that rely on an off-the-shelf captioning model, we design detailed techniques to improve the captioning quality. Most notably, we optimize the framework so that every character has a unique tag like "person-b" instead of general descriptions. As in Fig. 2, this enables downstream analysis to connect a character across different segments.

### 3.2.1 Map: Dense Scene Captioning

The Map step follows a sequence-parallel manner and generates dense captions, as in Fig. 2(a). It involves: **(1) Detailed Description.** We empirically discover that existing VLMs might struggle with processing video clips with significant transitions. Therefore, we first prompt VLMs to check the continuity of every short clip and specify the transitioning frame indexes; then, we conduct the captioning task by letting the VLMs describe every continuous *scene* in detail. Such a strategy decreases existing VLMs' difficulties and provides "scenes" as atomic long video understanding units. **(2) Key Characters and Objects.** As preparation of consistent character names, we sparsely sample frames from a longer video segment, instruct the VLM to identify the salient characters/objects, describe

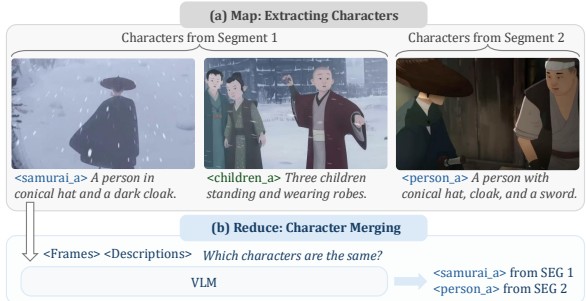

Figure 3: **Consistent Names.** (a) The Map step extracts the salient characters/objects along with a description. (b) Then, the Reduce step uses VLM to associate the repeated characters, enhancing the consistency.

their identifiable properties, and specify the frame indexes most saliently showing these characters/objects (as in Fig. 3(a)).

### 3.2.2 Reduce: Consistent Names

Identifying consistent characters is essential for understanding long videos. Otherwise, the analysis cannot capture the notion of "protagonist" as in the example of Fig. 2. However, a common challenge of existing VLMs is that they tend to provide a general description for a character, *e.g.*, a person, instead of referring to it using a consistent name across the long video. Therefore, our Reduce step overcomes this challenge by merging the key characters extracted from the Map step, as in Fig. 3(b).

Specifically, our Reduce step decouples this task into two sub-steps: *character association* and *caption modification*. (1) MR. Video instructs the VLM to associate the repeated characters/objects by observing the salient frames of extracted characters/objects, as in Fig. 3(b). (2) Then MR. Video assigns a new set of names for every character following the format of "<entity>_<index>" to avoid repeated names or losing semantic meanings. Finally, MR. Video accordingly updates the names in the original captions to the newly generated ones. Although using external tracking tools [8] might also be a valid solution, we use VLMs because of simplicity and the fact that videos' frequently changing scenes could break the assumption of trackers.

## 3.3 Analysis I: Question Intention Analysis

MR. Video emphasizes the importance of intention analysis because of the inherent *ambiguity of questions* in long-context understanding: the questions might only contain partial information, and the model has to recover crucial clues like "when," "how long," and "where" in the video to perceive. For example, Fig. 4 demonstrates multiple scenes potentially relevant to the questions, while only one should be correctly selected via reasoning. This stage utilizes the captions from the captioning stage (Sec. 3.2) and optionally includes video frames.

Compared with key-frame retrieval in previous video agents [8, 42] and scoring mechanisms in VLMs [12], MR. Video marks the importance of reasoning with global context to determine the relevant video segments, instead of purely relying on local video contents within short clips.

### 3.3.1 Map: Segment Intention Analysis

Without losing generality, we divide the video into non-overlapping short segments, each containing several atomic scenes split from the captioning stage. For an hour-long video with 1k scenes, we have approximately 30 segments. Then, the VLM processes the segments' aggregated captions and the middle frames of each scene to infer whether any scene provides helpful information for the question.

Within each segment, we instruct MR. Video to focus on "*what is the question asking about*" and generate a paragraph of analysis as in Fig. 2(b). Concretely, its response contains: (1) *Reasoning*: a paragraph analyzing the key subject/criteria mentioned by the questions and how the contents presented in the captions *could* align with the question in any perspective, *e.g.*, Fig. 4(c). (2) *Candidate Scenes*: the LLM then lists the potential scenes that could contribute to answering the question. Please note that this is distinct from directly

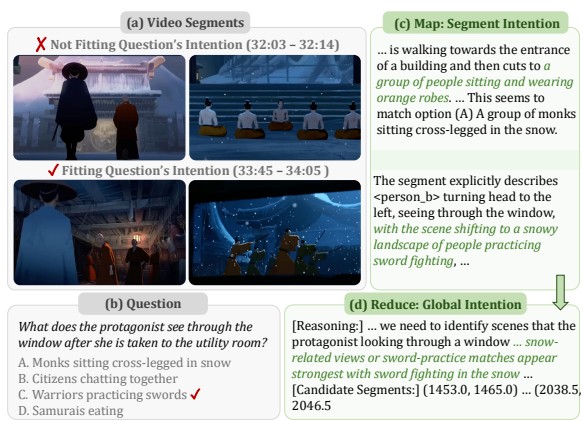

Figure 4: **Question Intention Analysis.** Long video questions require the model to recover the *hidden* information of the question, *e.g.*, who is "protagonist," what a "utility room" looks like. This motivates MR. Video's *explicit understanding of the question's intentions* by reasoning both the video contents and questions, instead of relying on conventional key-segment selection/retrieval.

retrieving key frames since it provides more contexts and allows frames that are helpful in *indirect* ways. (3) *Key Subjects*: The local caption segment becomes insufficient if the question mentions characters or criteria requiring global video information. So MR. Video specifies its unsure criteria and their identifiable properties here for the global Reduce step to analyze.

### 3.3.2 Reduce: Global Intention Analysis

MR. Video's Reduce step marks the key distinction with previous methods, where we reason the analyses at the video level. In principle, the Reduce step generates similar contents as the Map step but covers the contexts of the whole video. So it can localize the best scenes and subjects for the questions as in Fig. 2(b). The outputs contain the following contents. (1) *Reasoning*: a paragraph analyzing the key subject/criteria mentioned by the questions and how the contents presented in the captions *could* align with the question in any perspective, *e.g.*, Fig. 4(c); (2) *Candidate Scenes*: the LLM then lists the potential scenes that could contribute to answering the question. Please note that this is distinct from directly retrieving key frames since it provides more contexts. Fig. 4(d) shows an example output of the Reduce step, which correctly discovers the relevant video scenes by figuring out the protagonist and the window.

### 3.3.3 Key Segment Selection/Retrieval v.s. Our Intention Analysis

Explicitly reasoning the intention of questions, *i.e.*, completing the contexts, is a significant difference between our MapReduce principle and previous video agents [8, 42, 53]. We advocate for intention analysis, combining the whole video context instead of the key segment selection, which is a critical insight of MR. Video.

Our design uses the models' reasoning abilities to inspect short video clips in detail (Map) and then comprehend the video as a whole (Reduce). Although the key-segment selection of previous video agents [42, 53] and VLMs [12] implicitly reflects the "intention analysis" objective by choosing a few frames with the most similar features to the question, it is an over-simplified model for long contexts and reasoning: in the example of Fig. 4, it is challenging to extract features reflecting "protagonist," "utility room," or "windows" before understanding the video contexts. In addition, a sequential key frame selection framework assumes the small number of key events, which is not adaptive enough for complex or challenging queries, *e.g.*, the motivating "counting" example in Fig. 1.

### 3.4 Analysis II: Goal-Aware Analysis

Based on the analyzed question intentions, MR. Video's final MapReduce stage purposefully gathers the information related to the questions and converts them into a final answer, namely, "*goal-aware analysis*," as in Fig. 2(c). An essential functionality of this stage is that MR. Video should *explicitly plan the type of information* it needs: attending to captions and sparse frames over longer time horizons for reasoning, *e.g.*, Q2 in Fig. 2; or focusing on densely sampled frames benefits visual recognition, *e.g.*, Q1 in Fig. 2. With both capabilities, our MR. Video can flexibly handle a wide range of questions by aggregating the analysis.

### 3.4.1 Map: Goal-Aware Scene-centric Analysis

Starting from the candidate scenes generated by question intention analysis (Sec. 3.3), MR. Video proposes purposeful queries for VLMs to perceive *intra-segment* densely sampled frames for visual details or *inter-segment* sparsely sampled frames for global reasoning.

**Goal Proposal.** When generating the queries for VLMs, we are inspired by the flexibility of "Visual Programming" [11] and ViperGPT [36]: let LLM propose its queries for the VLMs and understand the candidate scenes in customized ways. As shown in Fig. 5, MR. Video proposes the VLM query to cover multiple aspects of the question, gathering comprehensive information.

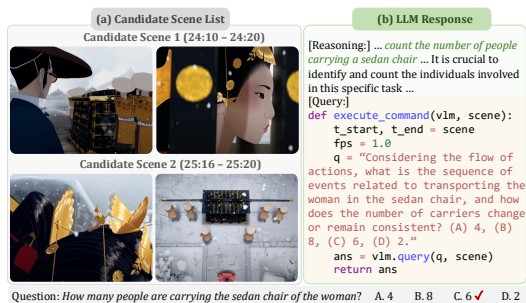

Figure 5: **Customized Queries for Perception.** With this question requiring detailed visual perception, MR. Video proposes goal-aware queries for the VLMs, confirming the criteria.

**Perceiving Local and Global Information.**
We employ the Map step as different strategies of sampling frames for VLMs to inspect. (1) *Local*: We densely sample frames within each short segment for objectives requiring detailed visual information, such as the example in Fig. 5. (2) *Global*: We sparsely sample frames across different segments, *e.g.*, the middle frames of the relevant segments identified by the intention analysis, and let VLM perceive them for information spanning longer temporal ranges. To better leverage the reasoning capabilities of LLMs, this step can also include the captions of the selected segments to simplify the perception.

### 3.4.2 Reduce: Answer Generation

The last Reduce step attends to the global context information and generates a final response. With the previous Map steps gradually summarizing the information, this Reduce step is no longer limited by context lengths and can fully unleash reasoning capabilities. As a notable characteristic of this Reduce step, it merges the scene analysis results together in a unified way, especially when different scenes provide contradictory perception results or require further calculation of scene-level information, such as counting queries.

### 3.5 Scalability Analysis

Beyond empirical accuracy, our MapReduce principle offers significant advantages from a systems and computational perspective. Here, we analyze the computational costs of **MR-Video** in comparison to sequence-to-sequence VLMs and other video agents.

**Premise: The Need for Comprehensive Context.** A meaningful comparison of computational cost must be grounded in the shared goal of comprehensive video understanding. As demonstrated in Fig. 1 and Fig. 2, tasks in long video reasoning often require a dense perception of the video's content to cover all the potential details. Therefore, our analysis is based on the assumption that all the methods are consuming the same amount of frames instead of deliberately reducing the amount of information (e.g., VLMs using sparse frames or agents selecting a few retrieved clips). Without the loss of generality, we compare the cost of our MapReduce with a conventional VLM or video agent when perceiving $N_{\text{frames}}$ frames.

**Comparison with Sequence-to-Sequence VLMs.** (1) **Token Count.** A standard VLM must process the visual tokens from all $N_{\text{frames}}$ frames, resulting in a total token count of $T_{\text{VLM}} = N_{\text{frames}} \times$ (tokens per frame). For our **MR-Video**, only the initial, parallelizable captioning step processes the raw frames, consuming approximately $2 \times T_{\text{VLM}}$ tokens (for two passes of segmentation and caption generation). Subsequent MapReduce stages operate primarily on text or sparsely sampled frames, incurring a much smaller average cost, $T_{\text{text}}$. Therefore, our approach expands the test-time computation in a targeted manner to build a comprehensive textual summary, which is more efficient for downstream reasoning. (2) **Computational Cost.** The advantage of MapReduce mostly lies in the computational cost. Even assuming that VLM can effectively understanding the $N_{\text{frames}}$ within its context lengths, a standard transformer-based VLM has a computational complexity that is quadratic with respect to the input length, i.e., $O(N_{\text{frames}}^2)$. In comparison, our MapReduce framework partitions the video into $M$ parallel segments. The computation is then reduced to $M \times O\left((N_{\text{frames}}/M)^2\right) = O(N_{\text{frames}}^2/M)$. Given that the number of segments $M$ for a long video is significantly greater than the number of 3 sequential stages in our framework, the total computational cost of **MR-Video** is substantially lower than that of a monolithic VLM attempting to process the same number of frames.

**Comparison with Video Agents.** Similar to the analysis of VLMs, we maintain the assumption that both methods start with a dense perception of $N_{\text{frames}}$ to generate high-quality captions or initial analyses. (1) **Token Count.** Under the dense context premise, both our method and video agents [8, 42, 45, 53] rely on an intensive initial captioning or analysis phase. Therefore, our total token counts are comparable to achieve the same quality of initial understanding. (2) **Critical Path and Parallelization.** The primary system-level advantage of our MapReduce principle is its ability to shorten the "critical path" of inference, enabling superior scalability. Consider the counting task in Fig. reffig:teaser, where over 50 key events of a soccer video must be identified. (a) A video agent relying on iterative, sequential key-frame retrieval would have a critical path of over 50 steps, with its length varying unpredictably based on video complexity and reasoning depth. (b) In contrast, **MR-Video** executes its plan using parallel "Map" steps, resulting in a short and controllable critical path of just 3 MapReduce stages. This inherent parallelism means that *the video processing throughput can scale linearly with the number of available GPUs or VLM inference endpoints*, a crucial advantage for practical deployment.

## 4 Experiments

### 4.1 Datasets

**Evaluation Dataset Selection.** To validate the MapReduce principle within our limited budget, we focus on the challenging long video benchmark: LVBench [39]. Compared with others [9, 31, 35, 63], LVBench features more extremely long video durations and challenging questions, as directly reflected by the lower accuracies of state-of-the-art models. With a limited budget, we expand the breadth of

| Model | ER | EU | KIR | TG | RE | SUM | Overall |
|---|---|---|---|---|---|---|---|
| *Proprietary VLMs* | | | | | | | |
| Gemini-1.5-Pro [37] | 32.1 | 30.9 | 39.3 | 31.8 | 27.0 | 32.8 | 33.1 |
| GPT4o [1] | 48.9 | 49.5 | 48.1 | 40.9 | 50.3 | **50.0** | 48.9 |
| Gemini-2.0-Flash [37] | 47.4 | 48.5 | 56.8 | 39.3 | 44.4 | 41.4 | 48.6 |
| *Open-sourced VLMs* | | | | | | | |
| InternVL2-40B [6] | 37.4 | 39.7 | 43.4 | 31.4 | 42.5 | 41.4 | 39.6 |
| TimeMarker [5] | 42.8 | 39.1 | 34.9 | 38.7 | 38.2 | 48.8 | 41.3 |
| Qwen2-VL-72B [38] | 38.0 | 41.1 | 38.3 | 41.4 | 46.5 | 46.6 | 41.3 |
| VideoLaMA3-2B [58] | 41.5 | 39.7 | 44.0 | 32.7 | 45.8 | 25.9 | 41.6 |
| mPLUG-Owl3 [56] | 46.0 | 41.6 | 42.4 | 41.1 | 47.5 | 40.4 | 43.5 |
| InternVL2.5-78B [6] | 43.8 | 42.0 | 42.1 | 36.8 | 51.0 | 37.9 | 43.6 |
| VideoLLaMA3-7B [59] | 45.8 | 42.4 | 47.8 | 35.9 | 45.8 | 36.2 | 45.3 |
| Qwen2.5-VL-72B [2] | - | - | - | - | - | - | 47.7 |
| ReTake [40] | 49.8 | 46.2 | 52.9 | 45.0 | 45.8 | 27.6 | 47.8 |
| VideoChat-Flash [22] | 51.1 | 46.0 | 49.0 | 38.9 | 48.5 | 34.5 | 48.2 |
| GLM-4V-Plus [10] | 46.2 | 47.8 | 54.1 | 42.7 | 46.5 | 37.9 | 48.7 |
| AdaReTaKe [41] | 53.0 | 50.7 | 62.2 | 45.5 | 54.7 | 37.9 | 53.3 |
| *Video Agents* | | | | | | | |
| VideoAgent [42] | 28.0 | 30.3 | 28.0 | 29.3 | 28.0 | 36.4 | 29.3 |
| VideoTree [45] | 30.3 | 25.1 | 26.5 | 27.7 | 31.9 | 25.5 | 28.8 |
| VCA [53] | 43.7 | 40.7 | 37.8 | 38.0 | 46.2 | 27.3 | 41.3 |
| MR. Video (Ours) | **59.8** | **57.4** | **71.4** | **58.8** | **57.7** | **50.0** | **60.8** |

Table 1: **LVBench Comparison.** Our MR. Video significantly outperforms previous methods by a large >7% margin, suggesting the effectiveness of the MapReduce principle. The VLM accuracies are from the official leaderboard as of 5/10/2025, and the video agent accuracies are from VCA [53]. The columns from "ER" to "SUM" represents different question types in LVBench, such as "entity recognition" and "summarization," details are in the supplementary materials.

evaluation using the subsets of other representative video understanding benchmarks, especially the long video parts of LongVideoBench [47], Video-MME [9], and EgoShema [31].

**Dataset Settings.** LVbench [39] curates 1,549 questions on 103 videos ranging from 30 min to 2 hrs, covering 6 video categories. We utilize the LVBench data as follows. **(a)** As of May 15th 2025, 4 out of 103 videos are unavailable from YouTube for downloading. So, our comparison in Sec. 4.3 utilizes all the remaining 1,492 questions. **(b)** For the ablation study (Sec. 4.4), we form a subset to save the budget by selecting the first video of each video category in LVBench. This subset has 6 videos and 98 questions in total. For additional evaluation, we use (1) the longest subset of LongVideoBench's validation set, (2) the long video subset of VideoMME without subtitles, and (3) the validation set of EgoSchema. The evaluation metrics are all accuracy for the benchmarks. More details on the datasets are in the Sec. D.4.

### 4.2 Implementation Details

**MR. Video Details.** Our MR. Video demonstrates a simple framework validating the MapReduce principle, only requiring one LLM for text understanding and one VLM for image understanding. To save our expenses, we utilize **Gemini-2.0-Flash** [37] as our VLM, and we only use GPT4o to process texts. On average, generating the dense captions for an hour-long video requires approximately $0.8 of Gemini-2.0-Flash, and answering each question from LVbench costs $0.4 GPT4o on average. We provide further details, especially the prompts, in Sec. D.

**Controlled Context Lengths.** We highlight a vital implementation detail so that our video agent is meaningful for overcoming the context length challenges: *we explicitly control the VLM to perceive less than 40 frames per query*, significantly less than the typical 256 or even more frames for long video VLMs [22]. This ensures MR. Video does not violate the motivation of building video agents.

**Baseline Evaluation.** Because of the high cost of evaluating models, we mainly refer to the numbers on the leaderboards or provided by the authors in our comparison (Table 1 and Table 2). The only exceptions are: (1) For VideoAgent [42] and VideoTree [45] on LongVideoBench (Table 2), we use their open-source code, the GPT4o model, and our captions for a fair comparison; (2) For our base VLM Gemini-2.0-Flash, we follow the standard VLM setting by uniformly sampling 256 video frames per video. More details are in Sec. D.5.1.

## 4.3 Main Comparison

### 4.3.1 LVBench Comparison

By tailoring long video understanding insights into the MapReduce principle, our MR. Video demonstrates a significant advantage on the challenging LVBench as in Table 1. Using the cheap Gemini-2.0-Flash and a smaller context length, our MR. Video improves the base VLMs and all the previous video agents primarily using a better GPT4o. Therefore, such a comparison suggests the effectiveness of our MR. Video for long video understanding.

### 4.3.2 Breadth Comparison

As shown in Table 2 (LVBench performance is listed for reference), our MR. Video demonstrates

significant advantages on the long video benchmarks than the previous video agents, despite using a cheaper VLM Gemini-Flash. MR. Video also *consistently outperforms the base VLM* with a smaller context length, while the previous video agents commonly underperform their VLM, GPT4o. Therefore, this indicates the effectiveness of MR. Video and the significance of the underlying MapReduce principle for long video understanding.

To guide the future analysis of video agents, we also notice the distinct question styles of LVBench, LongVideoBench, and VideoMME, leading to different scales of advantage between our video agent and the base VLM. Please refer to our discussion in the Sec. D.4.

| Benchmarks | LVBench Overall | LongVideoBench Val (Long) | EgoSchema Val | Video-MME Long (w/o Sub) |
|---|---|---|---|---|
| Average Duration | 4101s | 1434s | 180s | 2386s |
| *VLMs* | | | | |
| GPT4o [1] | 48.9 | 58.6 | 70.4 | **65.3** |
| Gemini-2.0-Flash [37] | 48.6 | 45.7 | 71.2 | 63.0 |
| *Video Agents* | | | | |
| VideoAgent [42] | 29.3 | 47.6 | 63.2 | 46.4 |
| VideoTree [45] | 28.8 | 39.2 | 67.0 | 53.1 |
| VCA [53] | 41.3 | - | 73.6 | 56.3 |
| MR. Video (Ours) | **60.8** | **61.6** | **73.8** | 63.4 |

Table 2: **Breadth Comparison.** MR. Video performs better than other video agents. More importantly, we consistently outperform the base VLM, Gemini-2.0-Flash, with a smaller context length, while other video agents commonly underperform their VLM, GPT4o. (LongVideoBench accuracy of GPT4o is from their paper, EgoSchema accuracies are from VCA [53], and Video-MME accuracies are from the official leaderboard and VCA's paper [53].)

## 4.4 Ablation Study and Analysis

We utilize the LVBench subset (explained in Sec. 4.1) to analyze our Map and Redyce operators.

**Consistent Character Names in Captions.** Following the order of the MapReduce steps, we first analyze the benefits of the Redyce step in captioning (Sec. 3.2): providing consistent characters/objects names. As shown in Fig. 2, such consistent names enable the analysis to capture coherent behaviors of characters. Without consistent names, we observe a significant performance drop ("w/o Consistent Characters" in Fig. 6).

**Question Intention Analysis.** As clarified in Sec. 3.3.3, we advocate comprehensively understanding the video contexts instead of the key-segment selection used by previous video agents. To analyze their differences, we utilize the target video clips annotated by LVBench to assess whether intention analysis can better localize the key segment: (1) whether the candidate scenes selected by our question intention analysis overlap with the annotated target clips; (2) whether the intention analysis is better than retrieving the key scene matching the embeddings of video clips and questions.

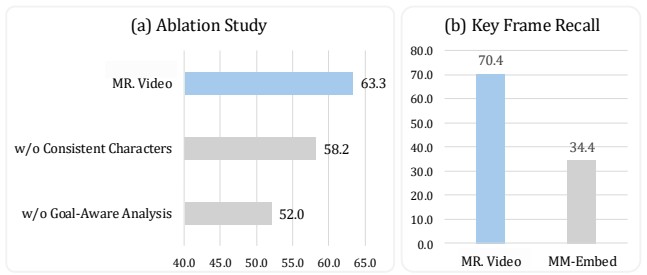

Figure 6: **Analysis**. (a) We investigate the benefits of MR. Video components. (b) The comparison between our question intention analysis and the key frame retrieval suggests the necessity of combining more video contexts for localizing the critical information.

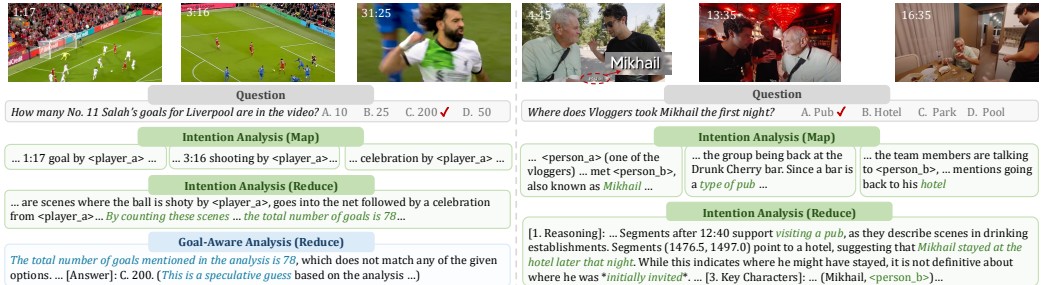

Figure 7: **Case Analysis. (Left)** Recalling the motivating example (Fig. 1), MR. Video checks every scene in detail (Map) and aggregates the whole video (Reduce). Although it misses some goals due to strict criteria (shooting, goal, and celebration), MR. Video shows the desired behavior of *counting exhaustively*. **(Right)** This example demonstrates how a consistent character name (Mikhail) benefits the reasoning process of MR. Video (first night).

First, MR. Video correctly localizes the relevant scenes for 70.4% of the questions, where a video typically contains 500-2k scenes. Second, we employ MM-Embed [25], a state-of-the-art multi-modal retrieval model, to conduct the key-frame retrieval. Under a fair comparison setup (details in the Sec. D.5.2), retrieval achieves an accuracy of 34.4%, which is significantly worse than our question intention analysis (Fig. 6(b)). This suggests the necessity of question intention analysis, which combines global context for localizing the key video segments.

**Goal-aware Analysis.** Goal-aware analysis (Sec. 3.4) provides the video agents with opportunities to delve deep into video content after coarse analysis. Without this final MapReduce stage, the performance drops significantly in "w/o Goal-Aware Analysis" (Fig. 6(a)).

## 4.5 Case Analysis

Finally, we closely observe the behavior of MR. Video and find it demonstrating a successful long video understanding process. **(1)** We recall our motivating problem of the challenging counting question: as in Fig. 7 (left), MR. Video indeed shows the behavior of exhaustively perceiving each video clip, checking the criterion, and summing up the numbers. Although its number is smaller than the ground truth due to strict checking criteria, MR. Video shows a valid path towards addressing a large number of events in long videos. **(2)** In this travel video, MR. Video demonstrates the multi-hop reasoning benefit from consistent names and explicit analysis of the event orders from global contexts, which are crucial premises for addressing complicated video reasoning.

## 5 Conclusion

To address the challenge of understanding both local details and global contexts in long video understanding, we introduce the MapReduce principle and formally define its operations in MR. Video. Compared with previous VLMs and video agents, MR. Video shows the advantage in smaller context length, better sequence-parallelism and inference scalability, and comprehensive global context understanding. Targeting the under-explored challenges of long videos, we further propose *consistent character names* in captions and *question intention analysis* to replace the conventional key frame retrieval. Finally, MR. Video achieves significant advantage on multiple long video benchmarks, showing the potential and effectiveness of MapReduce.

**Limitations and Future Work.** (1) We utilize the LLM agents paradigm because of its low cost, but the MapReduce principle is also conceptually compatible with VLMs, where local attention compresses short video segments and global attention at the final layers aggregates the global contexts. Therefore, a potential future work is to formulate and verify MapReduce for VLMs. (2) Another limitation of LLM agents is that LLMs are not aligned with the video understanding, especially when the texts used for visual reasoning could lose nuanced visual information (analysis in supplementary materials), so another future work is to conduct post-training for the LLMs of the video agents.

## Acknowledgments

This work was supported in part by NSF under Grants 2106825 and 2519216, the ONR Grant N00014-26-1-2099, the DARPA Young Faculty Award, the NIFA Award 2020-67021-32799, the Amazon-Illinois Center on AI for Interactive Conversational Experiences, the Toyota Research Institute, and the IBM-Illinois Discovery Accelerator Institute. This work used computational resources, including the NCSA Delta and DeltaAI supercomputers through allocations CIS230012, CIS240133, and CIS240387 from the Advanced Cyberinfrastructure Coordination Ecosystem: Services & Support (ACCESS) program, as well as the TACC Frontera supercomputer, Amazon Web Services (AWS), and OpenAI API through the National Artificial Intelligence Research Resource (NAIRR) Pilot.

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

## A    Delving into Long Video Benchmarks

In Sec. 4.3.2, we mentioned the fact that the long video datasets have nuanced differences in their preferred long video understanding capabilities, so the improvement and pattern of MR. Video displays different margins of advantage on these datasets. For instance, the improvement of MR. Video over the base VLM on LVBench [39] and LongVideoBench [47] is more significant than Video-MME [9], and such observations reveal the common challenges of video agents. In this section, we present several representative examples showing such distinctions of datasets and discuss the advantages and challenges of video agents. Please note that all of these benchmarks have curated a diverse set of questions. We demonstrate examples only to provide an intuition of the *complexity* of question styles instead of claiming that these benchmarks can be solved with a few techniques.

We show the examples in Fig. A, including the representative questions from LVBench [39], LongVideoBench [47], and Video-MME [9].

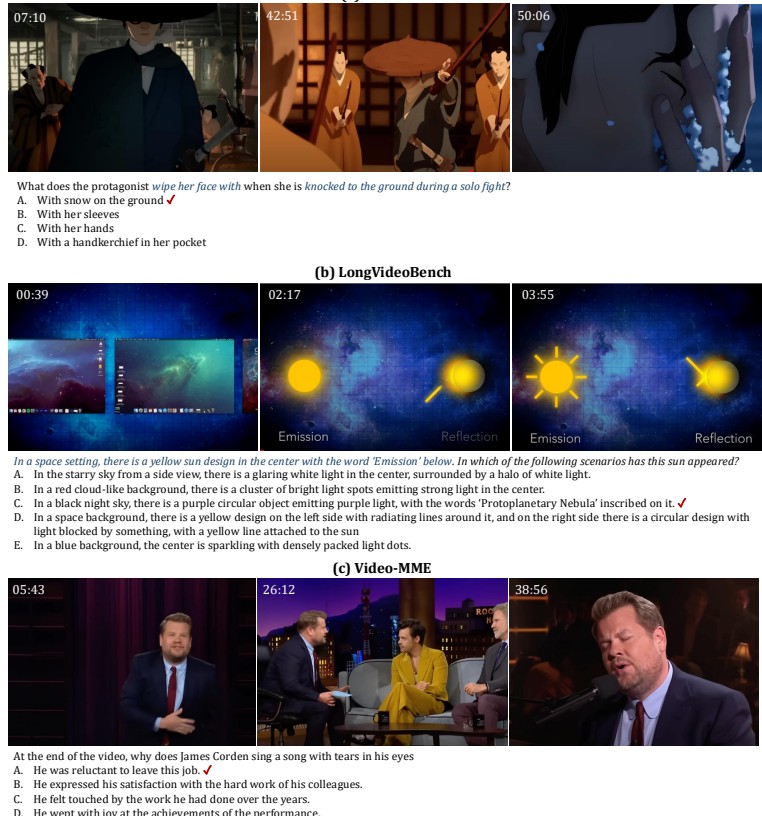

Figure A: **Examples of Different Long Video Benchmarks.** The benchmarks demonstrate different nuanced styles and desired capabilities from the long video understanding models. (a) LVBench [39] requires the model to precisely localize the key information by understanding the story, capturing the characters, *e.g.*, protagonist, and comprehending the question. (b) LongVideoBench [47] also emphasizes the importance of finding the key information, but the query is more explicit by directly naming the property to search for. (c) Video-MME [9] shows the questions closer to the style of interpretive queries, requiring the models to have a rough speculation and summarization of the video.

**LVBench.**    For LVBench, the model has to localize the scene of "solo fight" correctly and understand the meaning of "knock down" and "wipe face" to answer the question. Notably, the model has to integrate the contexts of the video and speculate the "protagonist" first to execute this task.

**LongVideoBench.**    Although both require precise localization, LongVideoBench is different from LVBench. LongVideoBench provides explicit and accurate visual cues for the model to localize the object, but the model has to propagate such information across the temporal axis to answer the

question. Compared with LVBench, LongVideoBench emphasizes models' visual detail perception and temporal association abilities.

**Video-MME.** Unlike the above two benchmarks, many questions in Video-MME are not about a specific event. Instead, they are more interpretative, similar to the impression of a human after watching the videos.

Comparatively, LVBench and LongVideoBench emphasize the challenges of localizing one or multiple key video clips and *exact* matching of contents, while Video-MME contains more *interpretative* questions similar to how humans gain an intuitive impression of a video segment. Even LVBench and LongVideoBench are slightly different: LongVideoBench provides more explicit vision-centric cues, and LVBench specifies more from a story or event aspect.

The improvement of our video agent over the base VLM, especially the improvement on LVBench and LongVideoBench, requiring the precise localization of information, demonstrates the advantage of video agents in localizing critical visual information by reasoning about the overall video context. Comparatively, video agents relying on text-based reasoning might lose visual details, making them less effective for interpretive questions like Video-MME. Even so, our MR. Video still outperforms the base VLM with a smaller context length, while all the other video agents fall behind their base VLMs. Therefore, the above analysis indicates the necessity of the MapReduce principle in handling a wide range of video tasks compared with other video agents.

# B    Scalability Analysis

In this section, we provide a detailed analysis of the token consumption of our **MR-Video** framework, accompanying Sec. 3.5. We compare our method with a representative open-source baseline, VideoAgent [42], on the LongVideoBench benchmark, following the setup in the main paper. This analysis reveals how the MapReduce principle intentionally utilizes more tokens to achieve a more comprehensive and reliable understanding of long videos, and how this design leads to superior inference-time scalability.

As shown in Table A, starting from identical video captions, our **MR-Video** consumes approximately 14x more tokens than VideoAgent to achieve a significantly higher accuracy. This substantial difference in token usage is not an incidental byproduct but a deliberate design choice central to our framework's philosophy. While VideoAgent restricts its agent to a maximum of 4 rounds of interaction with the video, our approach requires the agent to densely perceive the entire video content.

| Method | Avg Input Tokens Per QA | Avg Output Tokens Per QA | Accuracy (%) |
|---|---|---|---|
| VideoAgent [42] | 7,695 | 383 | 47.6 |
| **MR-Video (Ours)** | 109,522 | 4,908 | **61.6** |

Table A: Token consumption and accuracy comparison on LongVideoBench. Our MR-Video intentionally consumes more tokens to densely perceive the entire video, leading to significantly higher accuracy.

Such a contrast directly reflects the advantage of our design in densely perceiving the video, which is necessary (as explained in Sec. 3.5). More importantly, simply increasing the token budget does not trivially lead to better performance. In fact, VideoAgent's own ablation study (Fig. 3, left in their paper) suggests that increasing the number of perception rounds can cause performance to saturate or even decrease. This observation motivated our exploration of a new scaling paradigm. Instead of pursuing greater *depth* (more rounds of searching for key frames), our MapReduce principle improves the *breadth* of understanding by optimizing for information coverage.

At first glance, it may seem contradictory that a method requiring more tokens can offer better inference-time scalability. The key to understanding this is to analyze the *critical path* of computation when serving the model. Consider the event-counting scenario from Fig. 1, where over 50 goals must be identified. (1) **VideoAgent** relies on an iterative, sequential process of key-frame retrieval. Its critical path would consist of more than 50 sequential steps to identify all the key events, which grows

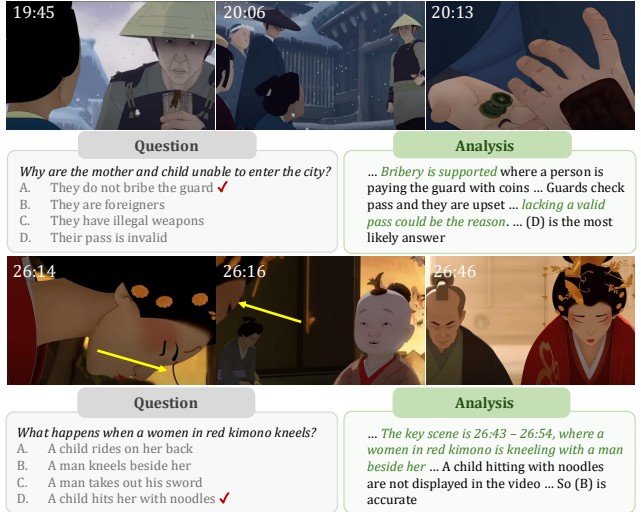

Figure B: **Failure Case Analysis.** MR. Video's failure largely comes from (1) the VLMs fail to *understand the narratives and scene transitions* (example 1); and (2) VLMs fail to capture visual details (example 2). (Noodles and the blurry face are pointed with arrows.)

with video complexity or reasoning hops. (2) **MR-Video**, however, relies on highly parallelizable "Map" steps. This results in a controllable critical path of just 3 MapReduce stages.

Consequently, our MapReduce principle is highly amenable to parallel computation. The system's throughput (i.e., the duration of video processed per unit of time) can scale linearly with the number of available GPUs or VLM inference endpoints. This design makes **MR-Video** exceptionally well-suited for practical, large-scale deployment where latency and throughput are critical.

## C Failure Case Analysis

As all the video agents utilize VLMs to interpret the video contents, the performance of the video agents is constrained by the underlying VLMs and LLMs, especially when the information relevant to the question is already localized successfully by the video agent. (1) The underlying VLM fails to capture the nuanced storyline in the example 1 of Fig. B: even though the model notices the bribery behaviors, it fails to conclude the correct answer due to not understanding the narrative of the videos. (2) Moreover, our video agent cannot recover the visual details overlooked by the VLM, such as the noodle and the woman's face in example 2 of Fig. B.

## D Prompts and Implementation Details

### D.1 Captioning Prompts

We describe the detailed steps and prompts for our dense captioning of the video (Sec. 3.2). All the datasets share the same captioning prompts.

**Map: Dense Scene Captioning.** As in Sec. 3.2, we let each short video segment produce its dense captioning, involving the following three map steps – all the video segments are independent within each step to support parallel inference:

1. We split each 10s video segment into individual scenes and check if the first scene of a segment can be merged with the last frame of the previous segment. Scene splitting prompts are in Table B, and the "Scene Merging" prompts are in Table C.

2. We identify the salient characters and use them to generate the dense captions in each video segment. The prompts are in Table D.

3. With the selected characters, we generate the dense captions of each scene with the prompts in Table E.

```
## Context
You will be given a few continuous screenshots of the video corresponding to approximately 10 seconds
of video duration, and provide detailed, faithful, and accurate analysis of this video segment.  The
objective of this analysis is to group the video into short segments based on the contents for the sake
of captioning and user question answering.

## Instructions
To perform the analysis of decomposing a video into shorter parts, let's do it step by step.
1.  Based on the provided frames of this video segment, please describe the contents of the video
segment briefly and accurately.  You should cover each action and event in the clip.  The description
should be detailed, faithful, and accurate.  It should come with a header:  "[1.  Description]:".
2.  Based on your description, please answer the following question:  "Is this video segment a single
scene or a combination of multiple scenes?" The definition of a scene is a single, self-contained, and
continuous event that could be easily summarized into one sentence by a human.  Your answer should come
with a header:  "[2.  Single:]"
3.  If the answer to the previous question is "no", please provide the index of frame(s) separating the
scenes from the given frame.  Your answer should come with a header:  "[3.  Frames]:" and in the format
of a list of integers.

## Example
Your response should be in the following format:
[1.  Description]:  This video shows ...
[2.  Single:  yes/no]:  No.
[3.  Frames]:  [5, 9]

Please pay special attention to:
- The precise localization of the frames is very important for downstream tasks.
- The summarization at the scene level should be consistent with the frames you provided.  For instance,
the number of scenes should be one more than the number of frames in the list.  If you provide 0 frames
since the images display a consistent scene, you will give 1 summary; If you provide 1 frame, there
should be 2 summaries; if you provide 2 frames, there should be 3 summaries, etc.
Now you will be presented the video frames, please perform the analysis carefully.
```

Table B: "Scene Splitting" prompts at the Captioning stage (Sec. D.1).

```
You are going to help with determining if a short video segment is a consistent scene.  You will be
given a few continuous screenshots of the video clip, and provide detailed, faithful, and accurate
analysis of this video segment.

Your objective is simple:  if *the video clip starting from the second frame* is a consistent scene
with *the first frame*.  Answer with "yes" or "no".

Now you will be presented the video frames, please perform the analysis carefully.
```

Table C: "Scene Merging" prompts at the Captioning stage (Sec. D.1).

**Reduce: Consistent Characters and Objects.**   As in Sec. 3.2, our additional "Reduce" step
enhances consistency by merging the repeated characters into unified names. It involves the following
steps:

1. We iteratively check if the characters from two video segments overlap with the prompts in
   Table F.
2. After assigning new names to all the characters/objects, we modify the old names in the
   original dense captions with the prompts of Table G.

### D.2   Analysis I Prompts

We describe the prompts for question intention analysis (Sec. 3.3).

**Map: Segment Intention Analysis.**   We let a standard LLM check the scene-level information and
understand the user intentions. Each chunk of captions contains 32 scenes. Its prompts are in Table H.

**Reduce: Global Intention Analysis.**   This step utilizes an LLM to process the segment-level
analyses from the previous step and unify them into a condensed video-level analysis. The prompts
are in Table I. The most critical part is explicitly instructing the LLM to conduct video-level reasoning
and find the most proper scenes.

### D.3   Analysis II Prompts

This section provides the details and prompts for MR. Video's goal-aware analysis (Sec. 3.4).

```
# 1. Motivation
You are paritipating in a video captioning task, but you can only watch a few frames of the video and
lack a broader context. Therefore, you will use using a character-centric and object-centric visual
memory that stores the key characters and objects in the video. Your objective is to identify the
potential key characters and objects from the video, that could be influential, and organize them into
a visual memory for downstream tasks.

# 2. Input and Output
You will be given the following inputs:
## 2.1 Input
You will have sveral sparsely sampled frames of the current video clip.
## 2.2 Output
Your output will have the following format:
[1. Appeared Characters]: You will return a list of the names of the characters and objects that
appeared in the current scene, from the visual memory. Strictly follows the format: [NAME1, NAME2,
...]
[2. Character Details]: You will return the details of the characters that appeared in the current
scene. Each item should contain the name of the character, a representative frame of the character,
and a description about how to identify the character in the frame. Format is:
[Visual Memory 1:] [[NAME: name], [DESCRIPTION: description], [FRAME: index of the selected frame to
display this character]] [Visual Memory Ends]
[Visual Memory 2:] [[NAME: name], [DESCRIPTION: description], [FRAME: index of the selected frame to
display this character]] [Visual Memory Ends] ...
Guidelines:
1. NAME should be a a general name, such as person_a, person_b, person_c, object_a, etc. Try to be
rigorous and faithful to the video without making assumptions.
2. DESCRIPTION should be a short description of the character's and object's appearance and properties,
especially how to uniquely identify the character or object from the representative frame.
3. FRAME should be the index of the frame that best represents the character or object in the scene,
favorably the most salient frame showing the front face of the character. It should start from 0.

## 2.3 Example Output:
[1. Appeared Characters]: ["person_a", "person_b", "dog_a"]
[2. Character Details]:
[Visual Memory 1:] [[NAME: person_b], [DESCRIPTION: a man with short hair and glasses in the frame],
[FRAME: 10]] [Visual Memory Ends]
[Visual Memory 2:] [[NAME: person_c], [DESCRIPTION: a woman with long hair and a blue dress in the
frame], [FRAME: 20]] [Visual Memory Ends]
[Visual Memory 3:] [[NAME: dog_a], [DESCRIPTION: a dog with standing beside the man with short hair],
[FRAME: 10]] [Visual Memory Ends]

# 3. Guidelines
This is not an easy task, please make sure to use your advanced reasoning ability and check every item
and step carefully. The following guidelines are very important for you to finish this task:
1. Please imagine yourself as a human watching the video, trying to perceive the salient things from
the video and understanding the deeper plots of the video.
2. When you are selecting the characters for the visual memory, please be picky:
(a) Only select the characters and objects that you believe are salient and could significantly
influence the plot. It could be a person in the movie, an animal in the documentary or cartoon, etc.
Make your best judgements.
(b) Only include a character if it is displayed saliently with great emphasis. Be conservative if you
cannot identify the character clearly. Better to be safe than sorry.
3. Format is very important. Please keep the strings in identical formattings to ensure smooth
post-processing.
4. Please make sure the [1. Appeared Characters] and [2. Character Details] are consistent.

# 4. Your Job
Now your job begins.
```

Table D: "Character Selection" prompts at the Captioning stage (Sec. D.1).

```
# 1.  Instructions
You will be given a few continuous screenshots of a video clip, some potential key characters and
objects of the video in your memory, and the caption of the previous scene.  Your objective is to
generate a caption for current displayed scene.  Your analysis will be faithful and accurate to the
video.
Input:
1.  Visual Memory:  the names, representative video frames, and the identifiable properties of the
characters and objects in the visual memory.
2.  Previous Caption:  the caption of the previous scene.
3.  Video Frames:  the few continuous screenshots of the current video clip.

# 2.  Guidelines
When generating the caption, please follow the guidelines below and solve this problem step by step:
1.  First, describe the main content of the current scene briefly.
2.  Second, use the visual memory to identify if any characters or objects from the visual memory
appear in the current scene.  If so, please list their name out.
3.  Third, describe the scene in detail, including the characters, their actions, the objects, the
properties of the characters and objects, the environment, and other types of contents, etc.
Some more detailed tips:
1.  When generating the captions, please take the previous scene as contexts and pretend that you
are watching a video continuously.  The goal is that a human should read your captions and feel like
watching a continuous video.
2.  When generating the captions, please be faithful to the video and make logical connections between
the scenes.
3.  When you encounter characters, please utilize the information and name from the visual memory if
what you see matches the visual memory.  For instance, if the visual memory contains a character named
"person_a", you should use <person_a> to refer to the character in your captions.
Important Rules:
1.  The quality of this step is very very very important.
2.  I want you to be very detailed and faithful to the video.  At least, you should go over the
following aspects:
2.1 What are the characters, what are their appearances, what are there clothes, what are their actions,
what are their emotions?
2.2 What are the objects, what are their properties, what are their relationships with the characters?
2.3 What are the environments, what are the background, what are the weather, what are the time of the
day?
2.4 Are there any text on the screen?  What are they?
2.5 If there is anything salient or anything weird, please describe it.

# 3.  Format
Your response should be in the following format:
[1.  Brief Description]:  ...  # captions, a string
[2.  Appeared Characters]:  ...  # the format of [NAME1, NAME2, ...], a list of character or object
names
[3.  Detailed Description]:  ...  # the detailed description of the scene, a string

# 4.  Your Job
Now your job begins.
```

Table E: "Dense Captioning" prompts at the Captioning stage (Sec. D.1).

**Map: Goal-aware Scene-centric Analysis.** Based on the information required to answer the question, MR. Video first proposes customized queries for each question as in Table J and applies these queries to the VLMs.

**Reduce: Answer Generation.** The final step is to combine the results of goal-aware scene-centric analysis with the global intention analysis to generate a final response. The prompts are in Table K.

## D.4 Datasets

**LVBench Videos.** We clarify the unavailable videos from LVBench, as mentioned in Sec. 4.1. LVBench requires users to download from YouTube with provided links to protect the copyright. As of March 1st, 2025, 4 videos are no longer available on YouTube, so we cannot evaluate them. Their IDs are: 28CIeC8cZks, idZkam9zqAs, QgWRyDV9Ozs, gXnhqF0TqqI. After filtering them out, we have 1,492 out of 1,543 questions. Therefore, MR. Video can still outperform the other methods by more than 5% even under the extreme assumption of counting the unavailable questions as "wrong" answers.

**LVBench Ablation Subset.** We select the first video of each category from LVBench (cartoon, live, self-media, documentary, TV, and sports) and form a subset for the ablation study, as mentioned in Sec. 4.1. Th six selected videos are: Cm73ma6Ibcs, TiQBTesZUJQ, t-RtDI2RWQs, hROKtPqktO8, rSE2YPcv89U, and CgvJqGxzRfE. They consist of 98 questions in total.

```
# 1.  Instructions
You will be given two sets of frames captured from a video, describing several characters or objects
from the video.  Your objective is to find if any character or object appears in both sets.  If so,
please help me locate the character or object and find the better frame representing the characters and
objects.
Input:
1.  Set 1:  the names, representative video frames, and the identifiable properties of the characters
and objects.
2.  Set 2:  the names, representative video frames, and the identifiable properties of the characters
and objects.

# 2.  Guidelines and Tips
This is not an easy task, please make sure to use your advanced reasoning ability and check every item
and step carefully.  The following guidelines are very important for you to finish this task:
1.  Please work on this problem via two steps:  (a) check if any items from the first set is repeated
with the second set; (b) if so, find the better frame representing the character or object.
2.  Please rely on both the video frame information and the identifiable properties to carefully
understand the characters and objects.
3.  When you are selecting the better frame for an object, please consider the following factors:  (a)
the frame should be the most salient frame showing the front face of the character; (b) the frame
should be the most representative frame showing the character or object.
4.  Sometimes the characters or objects are captured from different angles or distances, please make
your best judgement to check if they are the same character or object.

# 3.  Output Format
Please strictly follow the format below to ensure smooth post-processing:
[Repeated Characters and Objects]:  (Character_name1_in_Set_1, Character_name1_in_Set_2,
Better_character_name1), (Character_name2_in_Set_1, Character_name2_in_Set_2, Better_character_name2)
...
The answer lists all the repeated characters and objects in the two sets of frames, each tuple contains
three items describing the repeated character or object:
1.  Character_name_in_Set_1:  the name of the character or object in the first set of frames.
2.  Character_name_in_Set_2:  the name of the character or object in the second set of frames.
3.  Better_character_name:  the name of the better character or object that represents the
repeated character or object, must be consistent with the name in Character_name_in_Set_1 or
Character_name_in_Set_2.
An example output should be:
[Repeated Characters and Objects]:  (person_a, person_b, person_a), (dog_a, dog_b, dog_b)

# 4.  Your Job
Now you will receive two sets of frames and their character descriptions.  Please start your responses
with the information provided.
```

Table F: "Character Merging" prompts at the Captioning stage (Sec. D.1).

```
# 1.  Instructions
You will be given a description of a video clip, which potentially contains some characters.  After
some analysis, I have decided to change the name of the characters or objects, and your job is to help
me modify the descriptions to the new names.
Input:
1.  Old Description:  the old description of the video clip, containing the fields of Brief Description,
Appeared Characters, and Detailed Description.
2.  Modified List:  a list of characters to be modified in the format of OLD_NAME -> NEW Name.
Output:
Your output should be the modified description of the video clip strictly following the original format
and contents, only with names changed.

# 2.  Guidelines
1.  Only change the names, do not change the format or any contents.
2.  Please remember to update all the Brief Description, Appeared Characters, and Detailed Description.
3.  Keep the names consistent.
4.  The format of the characters in Brief and Detailed Description is <NAME>, please follow the same
format.

# 3.  Your Job
Now your job begins.
```

Table G: "Caption Modification" prompts at the Captioning stage (Sec. D.1).

```
# 1. Motivation
You will conduct the first step of long video understanding: **perceiving short video segments** and
**analyze their relevance to the user's question**.  By using short-segment analysis, you can avoid the
limitation of the model's context length for long videos.
You will have access to the following information for the current video segment:
1. A question.
2. The frames sampled from the video, each corresponding to a scene in the captions.
3. The captions of the video generated by a video captioning model, decomposed into short scenes
representing different video actions.  Notably, we have marked the potentially key characters or
objects using the format of <NAME>.  However, it is not entirely reliable (e.g., missing characters
or inconsistent tracking across frames), please use it with reasoning.

# 2. Output Formats

Please strictly following the output format below, which is important for post-processing.
[1. Reasoning]:  ...  (Your reasoning process.  Please be precise, concise, and clear.  Mentioning
evidence is any.)
[2. Relevant Segments]:  [(t_start, t_end), ...]...  (List the time range of the video segments that
are relevant to the question.  Please strictly follow the time information from the captions.  if you
think a continuous period is necessary for the question, merge them into a single segment.  Return an
empty list if none of the segments are relevant.)
[3. Confidence Level]:  ...  (Your confidence level.)
[4. Key Characters]:  [(character symnonym in question, identifiable properties or NAME in captions),
...]...  (The key characters that are mentioned in the question and how to identify them.  Keep the
list empty if the question is not related to any characters.)

# 3. Instructions and Guidelines
## Information Reliability
To principle is to **combine the information from the captions, video frames, and the question
(including the options, if any)** to analyze the user's intention.  The reliability of the information
is:
1. The question:  raised by the user, the most important and reliable.
2. Video frames:  reliable, but only covers a small portion of the video.
3. Captions:  less reliable, but covering more details, especially the "NAME" representing
character/object names.  You should combine the information from the question and video frames when
using the captions.

## Analysis Tips
1. Think carefully about how a short video segment could contribute to long video understanding by
paying attention to the question and video segment contents.  Some examples are:
- For question on visual details, you should check if the video segment **contains the scene that the
user wants**.
- For question on information over a period of time, such as the order or the number of actions, you
should reason **whether this segment can contribute part of the analysis**.
- For question on the reason or implication of the story/actions in the video, you should check if the
video segment **contains the key information** that can help you understand the story/actions.
2. Finding the key video segment is critical.  If the user mentions a clear criteria, such as specific
character of object, try to use it **precisely** and **rigorously** in your analysis.
3. If the question asks for certain characters in the plot/story, you should potentially localize its
NAME in the captions, or clearly specify its appearance properties.
4. Pay attention to the information reliability mentioned above.
5. Imagine yourself watching a video using the sampled frames and the captions.
6. When discussing your analysis, please provide the reasoning process and your confidence level
between 1 (almost guessing, no clear evidence of being relevant to the question) to 5 (almost certain,
clear evidence of being relevant to the question).
7. If the question should be answered with contexts, for "Relevant Segments", you should include
one more scene before and after the most possible scene to increase robustness.  For example, if the
most possible segment is (10, 20), and its previous and next scenes are (5, 10) and (20, 25), then you
should make it (5, 25) so that the contents between two scenes won't be missed.

## Your Input
1. The question:  a question coming with options.
2. The frames:  a list of frames sampled from the video.
3. The captions:  a list of captions decomposed into short scenes representing different video actions.
Each caption is the format of "(t_start, t_end):  caption".  Time is represented in seconds.

# 4. Your Job Starts
```

Table H: "Segment Intention Analysis" prompts at the Question Intention Analysis stage (Sec. D.2).

```
# 1.  Motivation
You will conduct **user intention analysis** as a step of long video understanding:  what is the
question asking about.  The questions from the users might be vague or not self-contained.  You will
complete the question by finding the relevant video segments, characters/objects, or how the short
video segments contribute to the long video understanding.
You will have access to the following information:
1.  A question.
2.  Your analysis of short video segments:  **is the video segment relevant to the question?**
Your analysis is the most important information in this step.  You will go through the analysis of each
segment containing the following parts:
1.  Reasoning:  ...  (your explanation)
2.  Relevant Segments:  [(t_start, t_end), ...]...  (The periods that are potentially relevant from
your analysis.  Time is represented in seconds.)
3.  Confidence Level:  ...  (Your confidence level.)
4.  Key Characters:  [(character symnonym in question, identifiable properties or NAME in captions),
...]...  (The key characters that are mentioned in the question and how to identify them.  Could be
unreliable.)

# 2.  Instructions and Guidelines
## Objectives
Your goal is to merge the information from separate short video segments into a complete understanding
at the video level.  Your most critical output for the downstream parts are the "relevant segments"
and "key characters".  Notably, you will carefully use your reasoning skills to handle the following
issues:
1.  Segment-level analysis might guess some relevant segments or characters for the question.  You
should select the most relevant segments and characters based on a video-level understanding, and
ignore the less relevant ones.
2.  Segment-level analysis might contain contradicting information since they come from separate
analyses.  You should carefully merge the information from different segments, and provide reliable
information for the downstream analyses steps.
3.  You should clarify how the results from segment-level can contribute to the long video
understanding.  For example, do we want to "sum", "merge", or "select" the information from individual
segments.

## Output Formats
[1. Reasoning]:  ...  (Your reasoning process.  Please be precise, concise, and clear.  Mentioning
evidence is any.)
[2. Relevant Segments]:  [(t_start, t_end), ...]...  (List the time range of the video segments that
are relevant to the question.  Please strictly follow the time information from the analysis provided
to you.  Merge the scenes if you think a continuous period is necessary for the question.)
[3. Key Characters]:  [(character symnonym in question, identifiable properties or NAME in captions),
...]...  (The key characters that are mentioned in the question and how to identify them.  Keep the
list empty if the question is not related to any characters.)
[4. Local or Global]:  ...  (Whether the question requires combining contexts from different segments
to answer.  If "yes", then this is a global question.  If "no", then this is a local question.)
It is very important to follow the format for the relevant segments section.  Every segment should be a
format of (t_start, t_end), especially the brackets should be "()" and matched.

## Principles and Tips
1.  Think carefully about how a short video segment could contribute to long video understanding by
paying attention to the question and video segment contents.  Some examples are:
- For question on visual details, you should check if the video segment **contains the scene that the
user wants**.
- For question on information over a period of time, such as the order or the number of actions, you
should reason **whether this segment can contribute part of the analysis**.
- For question on the reason or implication of the story/actions in the video, you should check if the
video segment **contains the key information** that can help you understand the story/actions.
2.  Finding the key video segment is critical.  If the user mentions a clear criteria, such as specific
character of object, try to use it **precisely** and **rigorously** in your analysis.
3.  If the question asks for certain characters in the plot/story, you should potentially localize its
<NAME> in the captions, or clearly specify its appearance properties.
4.  Imagine yourself watching a video using the sampled analysis.  Figuring out the flow of the plots
is critical.
5.  If the question is not really about the **whole video**, do not specify more than 10 relevant
segments.
6.  You should propose **at least 1 relevant segment**.  If you don't think any segment is relevant,
return a most likely segment and say "I have low confidence on the relevance of the segments".

# 3.  Your Job Starts
```

Table I: "Global Intention Analysis" prompts at the Question Intention Analysis stage (Sec. D.2).

```
# 1. Motivation
In this step of long video understanding, you are making preparations for calling vision-language
models to analyze sampled video frames.  Specifically, you will be given the user's question and a
video-level analysis from yourself.  Based on such information, you will **propose a question to prompt
the vision-language models** to analyze the video frames.
You will access the following information:
1.  A question.
2.  A video-level analysis from yourself.  It contains the following information:
1.  Reasoning:  ...  (Your explanation about which parts of the video are relevant to the question.)
2.  Relevant Segments:  [(t_start, t_end), ...]...  (The periods that are potentially relevant from
your analysis.  Time is represented in seconds.)
3.  Key Characters:  [(character symnonym in question, identifiable properties or NAME in captions),
...]...  (The key characters that are mentioned in the question and how to identify them.  Keep the
list empty if the question is not related to any characters.)
4.  Local or Global:  ...  (Whether the question requires combining contexts from different segments to
answer.  If "yes", then this is a global question.  If "no", then this is a local question.)

# 2.  Instructions and Guidelines
## Objectives
When thinking about the questions to ask, please pay attention to how the next step will sample the
video frames for your questions.  In practice, we will use two ways:
1.  Local:  Sample N video frames for each relevant segment, e.g., 32 frames.  In this way, the
vision-language models can use your question to check the details of each segment.
2.  Global:  Sample 1 video frame for each segment, sequentially.  In this way, the vision-language
models can use your question to check the flow of the plots or conduct reasoning over a longer period
of time.
Therefore, you should propose two questions:
1.  A local question:  what kind of detailed information or evidence should the vision-language models
find in each segment?
2.  A global question:  what kind of reasoning should the vision-language models conduct on a longer
time span?
## Output Formats
Please strictly follow the output formats below to propose your questions, so that the downstream parts
can easily extract the information:
[1.  Reasoning]:  ...  (Your reasoning process.  Please be precise, concise, and clear.  Explicitly
thinking about what kind of information is missing or important for the question.)
[2.  Local Question]:  ...  (Your question for the local analysis.)
[3.  Global Question]:  ...  (Your question for the global analysis.)
## Principles and Tips
1.  Think carefully about how a short video segment could contribute to long video understanding by
paying attention to the question and video segment contents.  Some examples are:
- For question on visual details, you should check if the video segment **contains the scene that the
user wants**.
- For question on information over a period of time, such as the order or the number of actions, you
should reason **whether this segment can contribute part of the analysis**.
- For question on the reason or implication of the story/actions in the video, you should check if the
video segment **contains the key information** that can help you understand the story/actions.
2.  Keep your question concise, clear, and within a few sentences.  Do not enumerate or explicitly
depending on any time information.
3.  Remember to use the options from the original questions, expressed with (A), (B), (C), (D), to
think about the best way to distinguish the correct one.  It is also important to include the original
options as the context for the vision-language models.
4.  Use your knowledge of prompting large language models or vision-language models to improve your
question.
5.  Your output questions should only contain a question and options.  Do not include any analyses,
speculations, or reasoning into the question.  For example, the question should directly start as
"Describe ...  (A) ..., (B) ..., (C) ..., (D) ..., (E) ...", "What is ...  (A) ..., (B) ..., (C) ...,
(D) ..., (E) ..." or similar formats.

# 3.  Your Job Starts
```

Table J: "Customized Queries for Perception" prompts at the Goal-aware Analysis stage (Sec. D.3).

```
# 1. Motivation
You are at the last step of long video understanding. You will have the user's question and a series
of your analysis to finally answer the user's question.
Before conducting actual analysis, it is important to understand the steps of the previoous analysis
that will be presented to you:
1. Video-level User Intention Analysis: You first analyze which parts of the video and what kind of
characters are relevant to the user's question. You also think about how each video segment could
contribute to the long video understanding.
2. Goal Proposal: To call vision-language models to analyze the video segments, you have proposed
two questions for the VLMs to use. The first question is called "local question", used for detailed
analysis for each segment, and the second question is called "global question", used for joint analysis
and reasoning across multiple segments.
3. Goal-aware Analysis: You will receive the results of the vision-language models' perception for
each video segment using the local question and across multiple segments using the global question.
By understanding the previous steps, you will have a good understanding of the meaning of the
information provided to you, especially which parts are reliable and informative for answering the
user's question.

# 2. Instructions and Guidelines
1. Think carefully about how a short video segment could contribute to long video understanding by
paying attention to the question and video segment contents. Some examples are:
- For question on visual details, you should check if the video segment **contains the scene that the
user wants**.
- For question on information over a period of time, such as the order or the number of actions, you
should reason **whether this segment can contribute part of the analysis**.
- For question on the reason or implication of the story/actions in the video, you should check if the
video segment **contains the key information** that can help you understand the story/actions.
2. Carefully consider whether the analysis at local segments or across multiple segments is more
important for answering the user's question.
3. With the information provided to you, imagine youself as a human watching the video. Figuring out
the flow of the plots is critical.
4. It is possible that some information is vague or contradicting each other. You should utilize
advanced reasoning skills to resolve the contradictions. Some very useful principles are:
- If the user has mentioned a specific criteria, try to use it **precisely** and **rigorously** in your
analysis.
- Try to utilize the confidence levels provided in the answers.
- Always thinking about your strategy: how the analysis at local segments or across multiple
segments could contribute to the long video understanding. For example, do you combine the pieces
of information together, summing some numbers, or picking the best segment to answer the question?
- Humans have a limited memory. Always prioritize the most salient information.
5. Pay attention to the time information. They might provide additional correspondence information
across different segments and analyses.

# 3. Output Format
Please provide your answer in the following format:
[1. Reasoning]: ... (Your advanced reasoning based on the information above.)
[2. Answer]: A capital letter from A to E (If you cannot find a correct answer, please make a guess
from A to E based on the information you have. To ensure correct post-processing, please strictly use
this format. Do not add any characters or spaces.)
# 4. Your Job Starts
----------------
```

Table K: "Answer Generation" prompts at the Goal-aware Analysis stage (Sec. D.3).

**Breadth Benchmarks.** As mentioned in Sec. 4.1, we utilize several long video benchmarks in addition to LVBench [39] to provide comprehensive evaluation. However, we evaluate on their subsets due to limited computation resources. (1) LongVideoBench [47]. We evaluate the official long video validation subset of LongVideoBench, containing videos with a duration of $(900, 3600]$ seconds. There are 188 videos and 564 questions in total. In the comparison, the accuracies of the VLMs come from Table 5 of the LongVideoBench paper. (2) EgoSchema [31]. We evaluate on the validation set of EgoSchema, which contains 500 videos and questions. The performance mainly comes from Table 2 of VCA [53]. (3) Video-MME [9]. Our evaluation follows the long video subset of Video-MME, under the setting of not using subtitles. This set contains 300 videos and 900 questions in total. The performance of models comes from the [3] as of March 1st 2025.

---

[3] https://video-mme.github.io/home_page.html#leaderboard

```
You are a helpful assistant with the ability of watching videos and answering the questions raised
by human users.  You will process a few continuous screenshots of the video, and answer the questions
raised by human users.  If you encounter any issues that you cannot answer the question, please pick
the most possible answer from the options.
When you answer, please follow the format of: [1. Reasoning]: ... (Why you choose this answer) [2.
Answer]: ... (The answer you choose, from A, B, C, D)
Important: If you cannot answer the question, please pick the most possible answer from A, B, C, D, E.
Do not leave it blank or select other options.
```

Table L: The prompts used for evaluating Gemini-2.0-Flash (Sec. D.5.1).

## D.5    Analytical Experiment Details

### D.5.1    Baseline Evaluation

As mentioned in Sec. 3.2, we evaluate Gemini-2.0-Flash on the long video benchmarks. For the 30min to hour ones, including LVBench, LongVideoBench, and Video-MME, we follow the standard setting of uniformly sampling 256 frames from each video. For EgoSchema, whose videos are 3min, we uniformly sample 128 frames for evaluation. With LVBench frequently asking about events of specific timestamps, we further provide each frame's seconds as interleaved images and texts. Since LongVideoBench's questions are commonly related to the subtitles, we provide the subtitles of sampled frames as the contexts to the VLM. The prompts used for evaluation are in Table L.

### D.5.2    Ablation Study on Finding Relevant Segments

This section describes more details about the analytical experiments conducted in Sec. 4.4, where we compare the question intention analysis of MR. Video with a multi-modal retriever, MM-Embed [25].

**Types of Questions.**    Since LVBench's annotations for "summarization" and "reasoning" questions might specify long ranges, our evaluation mainly focuses on the question types with precise intervals: key information retrieval, event understanding, entity recognition, and temporal grounding. On our subset for analysis, this results in 64 questions.

**MM-Embed's Retrieval.**    Following the practice of VideoAgent [42], every video frame is encoded by concatenating its image content with a timestamp since some questions are related to specific seconds. In addition, every question is encoded along with its multiple choices, as some questions do not contain specific contexts. Since MR. Video might propose multiple candidate scenes, we let the retriever select the same number of top-k candidates for a fair comparison. Finally, every question searches its relevant frames via the maximum inner product between the question and video frame embeddings.

## E    Broader Societal Impact

As a general principle for long video understanding and its corresponding agentic framework, our MR. Video does not introduce societal bias in this process. However, the MR. Video framework utilizes the VLMs and LLMs, represented by Gemini-2.0-Flash and GPT4o, so the results might indicate the societal biases of these models.

