# OpenReview forum: "MR. Video: MapReduce as an Effective Principle for Long Video Understanding"
_NeurIPS.cc/2025/Conference — NeurIPS 2025 poster_

### Official Review · Reviewer_wmFm · 2025-07-02

**Clarity:** 3
**Significance:** 3
**Originality:** 3
**Rating:** 4
**Confidence:** 4

**Summary:**

In this paper, the authors propose Mr. Video, a new video agent to realize long video understanding. It contains two part: map and reduce, with different prompts and strategies.

**Questions:**

Please refer to Strengths And Weaknesses

**Ethical Concerns:**

["NO or VERY MINOR ethics concerns only"]

**Final Justification:**

The rebuttal addressed my concerns, so I raise my score to 4.

**Limitations:**

Please refer to Strengths And Weaknesses

**Paper Formatting Concerns:**

NaN

**Quality:**

3

**Strengths And Weaknesses:**

### Strength
1. The paper is well-written
2. The results of the proposed method seem good

### Weakness
1. My main concern is fairness. Study of agent strategy with API models has a serious problem: the model varies with time, and we do not know the exact details. Such a black-box manner hinders the disentanglement of the performance between the improvement of the API itself and the effectiveness of the proposed agent strategy. So I think a study based on open-source SOTA models is necessary, for example, Qwen-2.5-VL-72B.

2. Following 1, the comparison with previous agents is also not fair. As mentioned in lines 302-307, the baselines are not re-evaluated with the same API. I understand the problem of evaluation cost. But I do not think it is a reason to conduct an unfair comparison.

3. I think the Consistent Name is an interesting and important part for an advanced agent. But Sec 3.2.2 is too brief for me to understand the exact solution to realize it. After I read the prompt from the supp. It seems it needs to check pairs of frame subsets for all the combinations to realize identity consistency; it seems costly. So it raises my concern about the reasonableness of the proposed solution.

Based on the above problem, despite the good benchmark results, I give a Borderline reject to the current paper. I will adjust it based on the comments from other reviewers and the rebuttal.

---

> ### Author Rebuttal · Authors · 2025-07-31
>
> We appreciate your insightful feedback and your recognition of the improvement we have achieved. We hope the following clarifications and additional experiments could address your concerns.
>
> ## 1. Proprietary Models for Video Agents
>
> > My main concern is fairness. Study of agent strategy with API models has a serious problem...
>
> We appreciate you raising this question and recognize our difficulty in controlling the expenses. We will first provide the essential background and move on to your deeper questions step by step.
>
> First, we use proprietary models following the **standard practice of existing video agents**, such as VCA [1], which is also compared in our paper.
>
> [1] Yang et al. VCA: Video Curious Agent for Long Video Understanding. ICCV 2025.
>
> Second, we clarify the confusion caused by the Gemini-2.0-Flash used in our paper and GPT4o used in previous works: **GPT4o is commonly considered as stronger than Gemini-Flash-2.0**. As in our Tables 1 and 2, GPT4o is better or similar to Gemini-2.0-Flash. In addition, GPT4o's cost is 250x higher than Gemini-2.0-Flash (2.5 dollars - 0.1 dollars per 1M input tokens), which also indicates that GPT4o is a generally stronger model. (More quantitative results are in the next section, according to your request.) Therefore,  our MR. Video is, in fact, based on a weaker VLM.
>
> ## 2. Fair Comparison with Gemini-2.0-Flash
>
> > As mentioned in lines 302-307, the baselines are not re-evaluated with the same API.
>
> We evaluated VideoTree and VideoAgent with GPT4o, following their original design choices in the paper, while holding the thought that “GPT4o is generally better than our Gemini-2.0-Flash” as mentioned above. To fulfill your request, we re-run both of them with Gemini-2.0-Flash on LongVideoBench (same setting as Table 2) for a fair comparison. From the results below, we can conclude that:
>
> * GPT4o is a stronger model in most cases than Gemini-2.0-Flash.
> * Our MR. Video has a significant advantage under this fair comparison.
>
> |                      | Acc (GPT4o) | Acc (Gemini-2.0-Flash) |
> | -------------------- | --------- | -------------------- |
> | Direct VLM Inference | 58.6      | 45.7                 |
> | VideoAgent           | 47.6      | 37.8                 |
> | VideoTree            | 39.2      | 41.6                 |
> | MR. Video (Ours)     |  -    | **61.6**                 |
>
> (We omit running our MR. Video with GPT4o as GPT4o is stronger than Gemini-2.0-Flash, and our limited expense during the rebuttal.)
>
> ## 3. Open Source Models
>
> > So I think a study based on open-source SOTA models is necessary, for example, Qwen-2.5-VL-72B.
>
> That’s a good suggestion. We add the experiments of using Qwen2.5-VL-72B as the VLM and Qwen2.5-72B as the LLM. Because no previous published video agents utilizing Qwen2.5-VL-72B on long videos, we mainly conduct an ablation experiment: *whether our agentic MR. Video can improve VLM*.
>
> In the experiments, we encounter an engineering challenge that inferring a 72B model on numerous video frames requires extensive GPU resources and expenses, so we utilize the APIs from OpenRouter, which commonly run stably with 32 frames. Accordingly, we compare (a) Qwen2.5-VL-72B uniformly sampling 32 frames from long videos, and (b) our MR. Video utilizing $\leq$8 frames per VLM inference, to show the relative effectiveness of our video agent framework. Finally, our MR. Video outperforms the VLM baseline by 45.9% over 38.8%. **This suggests that MR. Video is also applicable to open-source models.**
>
> To summarize our clarifications and attempts from the above sections:
>
> 1. We used proprietary models in consistency with the video agents' standard practice;
> 2. Our Gemini-2.0-Flash is commonly considered as weaker than the GPT4o used in other video agents;
> 3. When other agents use Gemini-2.0-Flash for a fair comparison, our MR. Video still demonstrates significant improvement;
> 4. The effectiveness of video agent frameworks can also be validated on the open-source Qwen model.
>
> ## 4. Cost of Consistent Naming
>
> We are encouraged that you also agree with the importance and interest of consistent naming, and appreciate the time you've spent with our detailed designs and prompts. Regarding your concerns about the cost of merging the character names across the video, let’s first recap the process of consistent naming and then analyze its computational cost in detail.
>
> Our consistent naming involves two steps:
>
> 1. We first let VLM propose salient characters and objects using sparse frames on long video segments (3min). This leads to approximately 20 sets of characters for a one-hour video. The prompt for this part is in Table C (supp).
> 2. Then we instruct the VLMs to merge the characters from different sets using the prompts in Table E (supp), which is the step you have efficiency concerns about.
>
> In character merging, our process is **different from** checking all the frame pairs in $\mathcal{O}(N^2)$, assuming $N$ character sets. Instead, we implement it to recursively **merge the adjacent two frame sets into one** until we have a single set of characters, which is $\mathcal{O}(N)$ VLM calls in total, and most of the steps can be run in parallel. With the total number of character sets being approximately 20 for an hour-long video, this cost is reasonable. A side note is that all the video agents generally consider captioning as pre-processing and can be reused on the same video for answering numerous questions, which further decreases the cost concern in this captioning step.
>
> As one of the first attempts at addressing name consistency in long video understanding, we hope the intuition above is reasonable for you. Our main design principle is to strike a balance between the two extremes of relying entirely on "detecting and tracking objects exhaustively," which causes significant computational burdens, and "only considering a few primary characters," which loses crucial information. We would be glad to learn if the reviewer has other feedback or suggestions.

---

> > ### Comment · Reviewer_wmFm · 2025-08-04
> >
> > The rebuttal addressed my concerns, so I raise my score to 4. Looking forward to the open source.

---

> > > ### Author Response · Authors · 2025-08-04
> > >
> > > Thank you so much for the insightful questions and feedback! We will incorporate your suggestions into our next version of the paper. We are glad to open-source the code if this paper gets accepted, and we are grateful and excited to receive your encouragement on this!

---

### Official Review · Reviewer_b2Lt · 2025-07-03

**Clarity:** 3
**Significance:** 3
**Originality:** 3
**Rating:** 4
**Confidence:** 3

**Summary:**

This paper introduces the MapReduce principle from big data processing to long video question answering. The authors build a pipeline including understanding, reasoning, and answering, and apply the principle to all three stages. In the first stage, the authors generate dense captions for videos and associate characters in multiple captions by using the same tokens. Then a VLM is prompted to analyze the relevance between questions and captions in the second stage. In the final stage, an LLM is instructed to gather additional information using the VLM and generate the final answers. The evaluation shows improvement over previous approaches and verifies the effectiveness of the three stages.

**Questions:**

1. The first step of the pipeline is segmenting videos. Does how to segment videos affect the following stages or the performance? How do the authors design segmentation? Does any important information break across segments?
2. Removing context length limits may be a huge benefit of this work, but in the Reduce stages, the model also needs to gather information across multiple video segments. Are the Reduce operations also subject to context lengths?
3. Could the authors provide some examples or even a quantitative study to show the advantages in Fig. 1(d)?  For example, the approach may need more computation but operates in parallel, so what is the overall cost?
4. Ablating character name consistency reduces performance less than I expected. Could we analyze why the framework can answer a large proportion of questions despite inconsistent naming?

Overall, I would be happy to raise the score if the questions can be properly addressed.

**Ethical Concerns:**

["NO or VERY MINOR ethics concerns only"]

**Final Justification:**

The authors respond to my concerns and questions thoroughly. I raise my score and recommend acceptance.

**Limitations:**

Yes

**Quality:**

2

**Strengths And Weaknesses:**

Strengths
1. The concept of MapReduce for processing long content is interesting and reasonable.
2. The proposed approach shows significant improvement in multiple categories and datasets.

Weaknesses
1. This work designs a dedicated test-time scaling tailored to long video question answering. It provides insight on how to properly understand videos and answer questions step-by-step, but the technical novelty may be limited.
2. The pipeline involves captioning and multistage analysis with a VLM and an LLM, which may require huge inference computation.
3. The authors highlight the advantages of MapReduce against seq-to-seq VLMs and video agents. However, the advantages are not verified in the evaluation.
4. (Minor) All figures contain a lot of small text, which might not be reader-friendly.

---

> ### Author Rebuttal · Authors · 2025-07-31
>
> We are thankful for your constructive feedback and suggestions! We truly appreciate your recognition of our MapReduce principle as "interesting and reasonable" and find our approach leading to "significant improvement in multiple categories and datasets." For your questions and concerns, we have prepared the following discussions:
>
> ## 1. Technical Novelty
>
> **Intellectual Contribution.** Our MR. Video **clarifies a critical myth in the previous video agents** and **conceptualizes the solution into a simple MapReduce principle**.
>
> 1. In contrast to the reliance on key frame retrieval in previous video agents, *e.g.*, VCA [1], we explicitly emphasize **the necessity of comprehensively understanding video contents**. We ground this idea with the examples of counting numerous events (Fig. 1) and the typical vagueness in long video questions (the identity of "protagonists" in Fig. 2), suggesting that video agents cannot make any assumptions about the questions and video contents.
> 2. Based on the above proposition of comprehensive understanding, we **conceptualize the MapReduce principle** to overcome the limitations of previous VLMs and video agents. (a) For VLMs, MapReduce is not limited by context lengths. (b) For video agents, MapReduce not only **improves video understanding via comprehensively covering all the segments** but also enables **better inference scalability from a system perspective**. Specifically, the critical path of MR. Video is shorter and more controllable than video agents relying on sequential steps. Considering the counting example in Fig. 1, previous video agents need >50 sequential steps to cover these events, while our MR. Video exhibits a controllable number of 3 MapReduce stages. (More explanations in the next section)
>
> **Method Contribution.** We also propose specific MapReduce steps with insights tailored to video understanding:
>
> 1. *Consistent character names* in the captioning stage, which is essential for multi-hop reasoning (ablation in Fig. 6, qualitative example in Fig. 7, right).
> 2. *Intention analysis* replaces key frame retrieval and more reliably identifies relevant video segments, as in the ablation study of Fig. 6, right.
>
> **Performance Improvement.** Finally, our proposed principles and designs demonstrate a **significant and non-trivial improvement** over the previous video agents, *e.g.*, >19% better than VCA [1] on LVBench.
>
> We hope the above clarifies our novelties, and we are also glad to find that the other reviewers find the "MapReduce process interesting and original" (Zpo3), and "MapReduce principle simple and general" and "consistent character/object names in the captions critical" (FVH9). We would be glad to discuss further if you have follow-up questions.
>
> [1] Yang et al. VCA: Video Curious Agent for Long Video Understanding. ICCV 2025.
>
> ## 2. Cost and Advantages of MapReduce
>
> Following the novelty discussion above, we clarify your two tightly connected questions.
>
> > The pipeline involves ... huge inference computation
>
> > … the advantages of MapReduce against seq-to-seq VLMs and video agents …
>
> First, we suggest that the evaluation (Tables 1 and 2) already highlights **the advantages of our MR. Video in terms of accuracy**. In addition to performance improvement, our MapReduce principle also demonstrates advantages from a system perspective, as demonstrated below.
>
> **Background: Necessity of Comprehensive Video Contexts**. Before the actual comparison, we would like to reiterate the key background (please kindly refer to the previous section for detailed discussion):  The necessity of comprehensive video understanding. Therefore, the comparison of computational costs is not meaningful if some methods deliberately digest incomplete contexts, *e.g.*, VLMs with sparse frames and video agents selecting a few retrieved frames.
>
> **Comparison with VLMs.** Bearing this background in mind, we assume MapReduce and VLMs are handling the videos at the same frame density to cover comprehensive information, *i.e.*, $N$ frames:
>
> 1. Token count. VLM's token count is equal to the visual tokens on the $N$ frames, denoted as $N\times K$. For our MR. Video, only the captioning step needs to watch the video frames densely, spending approximately $2 N\times K$ tokens (two passes of segmenting the videos and generating captions), while the other multi-stage steps use texts or sparsely sampled frames, only take $0.32 N\times K$ on average. Therefore, our token count can be treated as expanding the test-time computation for better comprehension.
> 2. Computation cost. Considering a common transformer-based VLM, it requires $\mathcal{O}(N^2)$ computation for $N$ frames, while our MapReduce splits this into $S$ parallel segments and decreases the computation to $\mathcal{O}(N^2/S)$. As the number of stages in MR. Video is smaller than the segments $S$ of a long video, and the total computational cost of our MR. Video is smaller than that of a VLM.
>
> **Comparison with Video Agents.** Similarly, we analyze the cost of existing video agents for $N$ frames:
>
> 1. Token count. Both our method and video agents rely on dense captioning, thus, they cost similar token counts as MR. Video to achieve the same quality of captions.
> 2. Critical path and parallelization. Our MapReduce principle enables better inference-time scalability and shows advantages in shorter "critical paths." Consider the counting task in Fig. 1, where a previous sequential video agent and our MR. Video both have to count > 50 key events: (a) Video agents relying on iterative key frame retrieval would take > 50 steps in their critical path, and the number of steps also varies with the complexity of videos and multi-hop reasoning, while (b) MR. Video relying on parallel map steps shows controllable critical paths of 3 MapReduce stages. As a result, **our MapReduce is advantageous in using parallel computation to enable inference-time scalability**, *i.e.*, **the duration of videos that could be processed in unit time can scale linearly with the number of GPUs** (or VLM inference bandwidths).
>
> To conclude, (1) our MapReduce is better in accuracy, and (2) a detailed comparison of computational costs better justifies our design philosophy and scalability from a system perspective compared to both seq-to-seq VLMs and previous video agents.
>
> ## 3. Segmenting Videos
>
> In our captioning, the first step is to segment the video into atomic scenes and conduct captioning for each scene (L141-152). The following paragraphs clarify your questions on this.
>
> **1. Motivation.** Existing VLMs might overlook details when the video has multiple transitions. Therefore, we segment videos into atomic scenes (or the concept of "shot" in films) to overcome these limitations of existing VLMs. Such an operation also saves redundant captioning, *e.g.*, a static scene of over 1min when people are talking.
>
> **2. Implementation.** We prompt VLMs to (1) decompose each video segment (*e.g.*, 10s) into atomic scenes according to transitions, then (2) for the scenes at the boundaries of 10s video segments, merge them if they show coherent actions. The prompts are in Tables A and B (supp).
>
> **3. Information Break.** To avoid such breaks, we (1) let the VLMs generate a brief caption during scene segmentation; and (2) use these captions from previous scenes as the contexts for dense captioning (Table D).
>
> **4. Influence of Scenes.** We find that scene segmentation influences the captioning quality and affects downstream performance. We add this ablation by direct dense captioning on 10s segments and running MR. Video: The accuracy drops from 63.3% to 60.2%, indicating the value of better designs in the captioning.
>
> ## 4. Context Limits for "Reduce"
>
> We clarify that VLMs are limited by context lengths in videos because of the large token number per image, which is commonly >200 (*i.e.*, for Gemini, each frame takes 258 tokens). Then, even sparsely sampling 128 frames can cost 32k contexts.
>
> In comparison, MR Video's reduce steps only operate on texts, which is more efficient. Concretely, "Global Intention Analysis" (Sec. 3.3.2) requires the longest contexts in our “reduce” steps by digesting all the segment-level analysis. Since an hour-long video contains approximately 30 segments (L189, main paper), this is equivalent to LLMs analyzing an essay with 30 paragraphs. Its average token count is 4.5k, while the maximum is 12k, both significantly smaller than the VLM's contexts mentioned above.  Therefore, our reduce step is less limited by context lengths.
>
> ## 5. Consistent Names
>
> > … answer a large proportion of questions despite inconsistent naming?
>
> This is a very insightful question, especially from a benchmark perspective: the majority of the questions from existing long video benchmarks can be answered or speculated without modeling the consistency of characters. Take the two questions in Fig. 2, for example:
>
> 1. "How many sticks does the protagonist put in the incense burner?" Even though the model does not identify "protagonist," it can still correctly answer the question by finding the only scene with "incense burner."
> 2. "Why does the protagonist tie iron to her limbs?" This question seems to require the identification of the protagonist, but the LLM can utilize its knowledge of “tying iron” to speculate that this action is related to fitness.
>
> Despite benchmark limitations, we believe that consistent characters are essential for human-like video understanding and have shown their effectiveness. Showcasing its importance requires better benchmarks in future work to avoid the shortcuts of VLMs. We hope our demonstration of the benefits in the ablation study (Fig. 6) and quantitative observation in Fig. 7 (right), where the identification of a specific character "Mikhail" benefits multi-hop reasoning, and initiates attention into this domain.
>
> ## 6. Texts in Figures
>
> Thanks for the feedback, and we will fix this in the revisions following your advice.

---

> > ### Comment · Reviewer_b2Lt · 2025-08-04
> > **Thank you for the response**
> >
> > My concerns and questions have been addressed thoroughly. I will raise my score.

---

> > > ### Author Response · Authors · 2025-08-04
> > >
> > > Thank you so much for the insightful feedback! We truly appreciate your consideration and will incorporate your suggestions in the next version of our paper!

---

### Official Review · Reviewer_FvH9 · 2025-07-03

**Clarity:** 3
**Significance:** 3
**Originality:** 3
**Rating:** 4
**Confidence:** 4

**Summary:**

MR. Video introduces a MapReduce-inspired framework to handle long video understanding. It breaks a video into short, densely captioned clips ("Map") and then jointly aggregates their information ("Reduce") for deeper reasoning. It explicitly makes characters consistent across different scenes for coherence. On the LVBench benchmark of hour-long videos, MR. Video boosts accuracy significantly, reaching 60.8% accuracy.

**Questions:**

1. Given the widely adopted thought that inference-time scaling improves accuracy, it would be better if the authors could provide the average number of tokens being used for all the methods. This is particularly crucial when comparing agent-based methods, as their system prompts may vary significantly. This may provide a more comprehensive comparison.

2. It would be interesting to see how the reasoning model would work in this setup. I'm just curious about the performance. The result being better or worse will not affect my final scoring.

**Ethical Concerns:**

["NO or VERY MINOR ethics concerns only"]

**Final Justification:**

I appreciate the authors for providing the rebuttal. My concerns are addressed. I particularly like the analysis on "3 Token counts". I suggest adding the discussion in the final revision, in which case, you will have more time to analyze other baselines beyond VideoAgent.

**Limitations:**

Yes.

**Quality:**

3

**Strengths And Weaknesses:**

## Strengths

1. The Map-Reduce principle is simple and general for long video understanding.

2. The consistent character/object names in the captions are critical. It also benefits the reasoning of actions and stories across long horizons.

3. Clear improvement over previous methods on LVBench.

## Weaknesses

1. The character name takes the template of “<entity>_<index>”. I am curious how the set of entities is built and which entity is chosen. Using Figure 3 as an example, my question would be why the character in Scene 1 Shot 1 has to be <samurai_a> rather than <people_a>?

2. The paper has not explored to what extent character consistency persists. What if the character changes appearance (due to aging) or their attire (due to different plots)?.

---

> ### Author Rebuttal · Authors · 2025-07-31
>
> Thank you for your positive feedback! We are glad to know that you find our MapReduce principle simple and general for long video understanding, our improvement significant, and that several other designs sound beneficial to you. Regarding your concerns and questions, we clarify as follows:
>
> ## 1. Design of Character Names
>
> Our key intuition to achieving human-like character understanding is to recognize all the salient objects and characters that could catch a human’s attention. To achieve this, we design the following steps:
>
> > how the set of entities is built and which entity is chosen
>
> The entities are built by instructing the VLMs with our prompts (Table C) designed to identify the salient characters and explicitly distinguishing them via a special naming convention. The three crucial steps are:
>
> 1. Sparsely sample video frames (L155), specifically, 30 frames for each 3min, for the video to identify salient characters and objects;
> 2. Instruct the VLMs to "identify the potential key characters and objects from the video" (Prompts in Table C);
> 3. Provides in-context examples to explicitly ask VLMs to name the characters "in general names, such as person_a, person_b" (Prompts in Table C).
>
> > Using Figure 3 as an example, my question would be why the character in Scene 1 Shot 1 has to be [object Object] rather than [object Object]
>
> Based on the above discussion, we design the human-readable format of `[{object}_{index}]` for the VLM to follow, *i.e.*, the ability to generate names like < samurai\_a> mainly comes from our in-context examples and VLM's predictions.
>
> ## 2. Persistence of Character Names
>
> This is indeed one of the most challenging problems in long video understanding, but still does not have an established solution or analytical framework, to the best of our knowledge. Therefore, we focus on providing an empirical discussion for your questions. We will also gladly include these discussions in the revisions of the paper.
>
> Since we are using VLMs to conduct consistent naming, the performance is primarily bottlenecked by the capabilities of existing VLMs, *e.g.*, Gemini-2.0-Flash, and will improve in the future. Even so, we notice that:
>
> 1. The VLMs are good at identifying celebrities or famous characters consistently over a long sequence of time, *e.g.*, the player Salah is consistently recognized throughout the video in the example of Fig. 7 (left).
> 2. For less well-known characters, VLMs primarily rely on facial features to conduct the matching. For example, we checked the example video from Fig. 7 (right), specifically the character of "Mikhail." We found that the identification of Mikhail was consistent throughout the video, including the later video parts when his clothes were different. (The latter parts of this video were not displayed in the main paper due to space limits. To incorporate this discussion, we will include more frames of this video in our revisions.)
>
> Despite observing successful cases, we acknowledge that identifying consistent characters remains challenging in long videos for existing VLMs. Concretely, we check the captions for the video shown in Fig. 2, where the main character has demonstrated multiple ages (childhood and adult), clothing, and camera angles. Although our pipeline manages to merge some of them, VLM struggles to realize that some scenes are different stages of the same character.
>
> From this aspect, we hope our study can initiate the investigation into long-term character consistency by illustrating its utility in video QA (Fig. 6), so that this problem can be better noticed and addressed in future works.
>
> ## 3. Token Counts
>
> According to your suggestion, we analyzed the average token count comparison. Because of limited resources and time during the rebuttal, we primarily compare our MR. Video and the representative open-source VideoAgent [1] on LongVideoBench, following the setup of Table 2. We found that your insightful question is a unique angle to demonstrate the distinctions and advantages of our MapReduce principle compared with previous video agents.
>
> [1] Wang et al. VideoAgent: Long-form Video Understanding with Large Language Model as Agent. ECCV 2024.
>
> **3.1 Token Count Comparison**
>
> Beginning from the same captions, our MR. Video consumes approximately 14x more tokens than VideoAgent to achieve a significantly better accuracy, as shown in the table below. Such a difference comes from the fact that:
> * VideoAgent only allows the agent to interact with the video for at most four rounds, while
> * our MR. Video requires the agent to densely perceive the whole video.
>
> Then a potential follow-up question arises: **Is such a design of using more tokens to achieve better accuracy valid?**
>
> |                  | Avg Input Tokens Per QA | Avg Output Tokens Per QA | Accuracy |
> | ---------------- | ----------------------- | ------------------------ | ---------------- |
> | VideoAgent       | 7695                    | 383                      | 47.6\%             |
> | MR. Video (Ours) | 109522                   | 4908                     | 61.6\%             |
>
> For MR. Video, such behavior is deliberate to achieve more reliable video understanding, and this is also the major distinction between our MapReduce principle and the previous video agents: **the agent is necessary to utilize more tokens and conduct a comprehensive analysis of the video**. In the paragraphs below, we explain this comparison step by step via three questions.
>
> * Why and how MR. Video aims at utilizing more tokens for more comprehensive analysis?
> * Is it trivial to improve video understanding by simply using more tokens?
> * Why do we claim the MapReduce principle to support better inference-time scaling while it requires more tokens?
>
>
> **3.2 Necessity of Using More Tokens for Comprehensive Understanding**
>
> VideoAgent commonly relies on key frame retrieval, where it checks the video for an upper-bound of 4 rounds, while our MR. Video emphasizes intention analysis, where a VLM has to understand the whole video content comprehensively. This is the primary stage where our MR. Video utilizes more tokens. Such a design stems from our key insight: **the necessity of a comprehensive understanding of the video, without the reliance on key frame retrieval**. In the paper, we ground this proposition with the examples of counting numerous events (Fig. 1) and the common vagueness in long video questions (the identity of "protagonists" in Fig. 2), suggesting that video agents cannot make any assumptions about the questions and video contents.
>
> **3.3 Non-trivial Test-time Scaling**
>
> Under the principle of comprehensive video understanding and aiming at the benefits of test-time scaling, it is non-trivial to directly improve accuracy by simply spending more tokens. For example, Fig. 3 (left) of VideoAgent suggests that **increasing the round of video perception would saturate or even decrease the accuracy**. Such an observation motivated us to explore a new paradigm of test-time scaling for long video understanding: instead of pursuing the direction of **depth** in previous video agents, *i.e.*, increasing the rounds of searching for key frames, our MapReduce principle introduces another direction of improving the **breadth** of understanding, *i.e.*, optimizing for the coverage of information. Driven by this new direction, our MR. Video converts the additional tokens into covering every part of the video and effectively leads to an improvement in accuracy.
>
> **3.4 MR. Video's Advantages in Inference Scalability**
>
> Finally, we clarify the seemingly contradictory point: how can MR. Video be more friendly for inference-time scaling while utilizing more tokens? The key insight is to consider the **critical path** of answering a user's question correctly when we serve a video agent framework on systems.
>
> Take the scenario in Fig. 1, for example, where both VideoAgent and our MR. Video need to count > 50 events of goals: (1) VideoAgent relies on iterative key frame retrieval, so its critical path is more than 50 steps and could be even longer for more complex videos or multi-hop reasoning; however, (2) our MR. Video mainly relies on parallel steps (“Map” in MapReduce) and shows a controllable critical path of 3 MapReduce stages. As a result, **our MapReduce principle is advantageous in using parallel computation to enable inference-time scalability**, *i.e.*, **the duration of videos that could be processed in unit time can scale linearly with the number of GPUs** (or VLM inference bandwidths) under our design.
>
> **To conclude**, we hope the above discussion both answers the reviewer's inquiry on token counts and clarifies how this marks the distinctions and advantages of our MapReduce principle for test-time scaling. We would be glad to discuss with the reviewer for any follow-up questions.
>
> ## 4. Reasoning Models
>
> Thanks for the suggestion! We shift our previous GPT4o to Gemini-2.5-Flash (with thinking ability) for handling text processing in MR. Video (note: our vision component still uses Gemini-2.0-Flash, which does not have thinking abilities). We experiment following our ablation study setting (Sec. 4.4), and the accuracy improves from 63.3% to 66.3%. Therefore, **reasoning model can benefit our MR. Video**.

---

### Official Review · Reviewer_Zpo3 · 2025-07-05

**Clarity:** 3
**Significance:** 3
**Originality:** 3
**Rating:** 4
**Confidence:** 4

**Summary:**

This paper introduces MR. Video, a novel long video understanding framework inspired by the MapReduce paradigm. The system processes video in two MapReduce stages: one for generating consistent, detailed captions and another for question-specific reasoning. It tackles the core challenge of jointly handling dense local perception and comprehensive global reasoning under limited context length constraints. Experiments on LVBench dataset show improvement over previous methods.

**Questions:**

- While the authors acknowledge using GPT-4o and Gemini-Flash for cost control, so how is the performance of proposed approach with fully open-source models (e.g., Qwen, InternVL) instead of proprietary models?

- The main comparisons rely on different model configurations (e.g., MR. Video uses Gemini and GPT-4o, while other, like in VideoAgent, use GPT-4o alone). Could the authors clarify how they ensured fair comparisons? For example, are the gains solely attributable to better orchestration or also due to stronger base models?

- The ablation study reports gains from consistent character naming and intention analysis. Could the authors elaborate further on how robust these components are across videos with high character diversity or minimal dialogue?

**Ethical Concerns:**

["NO or VERY MINOR ethics concerns only"]

**Final Justification:**

As noted in my comments on the rebuttal, I decided to maintain my positive rating on the contribution of the paper.

**Limitations:**

Yes

**Quality:**

3

**Strengths And Weaknesses:**

***Strengths***
-	The reformulation of long video understanding as a MapReduce process is particularly interesting and original.
-	The paper presents two concrete Map and Reduce stages with clear definitions for each step, making it generally easy to follow.

***Weaknesses***
- MR. Video uses proprietary models including GPT-4o and Gemini-Flash, which would limit reproducibility. Besides, I also concern about evaluation fairness against  other baselines using different LLM/VLM combinations.
- There is no discussions of error propagation (e.g., how errors in captioning affect downstream analysis). This could raise a concern given the reliance on intermediate generated text.

 - The qualitative examples used in the paper appear to be animated or stylized content (e.g., cartoons or anime) rather than real-world video data. It raises concerns about the generalizability of MR. Video to other data domain like real, unconstrained, or egocentric video settings, where visual noise, occlusion, and ambiguity are more prominent.

---

> ### Author Rebuttal · Authors · 2025-07-31
>
> We are grateful for your positive feedback! We are encouraged that you recognize the simplicity and effectiveness of our MapReduce design for video agents. We appreciate your constructive questions and will clarify them below.
>
> ## 1. Proprietary Models & Fair Comparison
>
> **Common Practice of Using Proprietary Models in Video Agents**. We appreciate your raising this question and recognizing our challenges in expense control. The main reason we use proprietary models is to follow the common practice in most existing video agents, such as VCA [1].
>
> [1] Yang et al. VCA: Video Curious Agent for Long Video Understanding. ICCV 2025.
>
> **Difference between Gemini-2.0-Flash and GPT4o.** Second, we clarify the difference between Gemini-2.0-Flash used in our paper and GPT4o used in video agents, *e.g.*, VCA, regarding fair comparison. GPT4o is commonly considered stronger than Gemini-Flash-2.0: In our Tables 1 and 2, GPT4o is better or has similar performance to Gemini-2.0-Flash. In addition, GPT4o's cost is 250x higher than Gemini-2.0-Flash (2.5 dollars v.s. 0.1 dollars per 1M input tokens), which also indicates that GPT4o is a generally stronger model.
>
> **Comparison: Other Video Agents in Gemini-2.0-Flash.** We further add the experiments of evaluating the previous video agents with Gemini-2.0-Flash instead of GPT4o as an example of fair comparison. We mainly include VideoAgent and VideoTree because they have open-sourced their code. We conduct this on LongVideoBench, identical to the experiments shown in Table 2. As shown in the table below, our MR. Video still outperforms previous video agents after switching their VLMs to Gemini-2.0-Flash for a fair comparison. (We omit running our MR. Video with GPT4o as GPT4o is stronger, and our limited expense and time during the rebuttal.)
>
> | Method               | Acc (GPT4o)             | Acc (Gemini-2.0-Flash) |
> |----------------------|------------------------|------------------------|
> | Direct VLM Inference | 58.6                   | 45.7                   |
> | VideoAgent           | 47.6                   | 37.8                   |
> | VideoTree            | 39.2                   | 42.6                   |
> | MR. Video (Ours)     | -     | **61.6**                   |
>
> **Open-source models.** Third, we add the suggested experiments of using open-source VLMs and LLMs: Qwen2.5-VL-72B as the VLM, and Qwen2.5-72B as the LLM. We follow the same setting of our ablation experiments (Sec. 4.4). An engineering issue we encountered was the high GPU demand and price of running the 72B model on numerous frames. So we utilize the APIs from OpenRouter, which runs stably with less than 32 frames. In the comparison, we run the baseline of Qwen2.5-VL-72B model for direct inference using 32 frames, and let our MR. Video to utilize $\leq$8 frames to show the effectiveness of using shorter context lengths. Finally, our MR. Video outperforms the Qwen2.5-VL-72B baseline by 45.9% over 38.8%. This suggests that our framework is also applicable to open-source models.
>
> Therefore, our use of proprietary models is (1) consistent with the video agent's standard practice; (2) utilizing a weaker Gemini-2.0-Flash as the VLM; and (3) can also improve open-source Qwen models.
>
> ## 2. Analysis of Captioning's Error Propagation
>
> The error propagation from captions to the final results is an interesting and important topic. Unfortunately, this problem still does not have an established analytical framework, to the best of our knowledge. Therefore, we provide the following empirical observations:
>
> 1. Quantitatively, our ablation study in Fig. 6 analyzes "what if the captions do not model consistent characters?" The drop in accuracy (63.3% to 58.2%) suggests the influence of caption quality on the downstream analysis: if the caption quality decreases, the final question-answering accuracy will be impacted.
> 2. Qualitatively, we observe that the subsequent video agent components will fail if the captions overlook crucial information. In example 2 of Fig. B (supp), we discussed the captions overlooking the details of a key frame (noodles on the woman's face), which caused this frame to be ignored by the later stage of the MR. Video.
> 3. Our design also implicitly alleviates the error propagation from captions via incorporating raw visual information: specifically, our segment intention analysis (Sec. 3.3.1) sparsely samples the middle frame of each scene (L192, main paper) to assist the localization of relevant scenes without entirely relying on the captions.
>
> ## 3. Real-world Videos & Generalization
>
> Thank you for looking at our examples closely!
>
> **Our MR. Video can generalize to a wide range of videos** (real and unconstrained video settings). Concretely, the major benchmark LVBench (Table 1) covers diverse videos beyond anime, such as live TV, documentaries, etc. Moreover, we also evaluate on a broad range of benchmarks (Table 2), including the datasets of LongVideoBench, VideoMME, and EgoSchema, where our MR. Video achieves a consistent advantage. These suggest the generalization of our framework.
>
> Regarding "qualitative examples ... appear to be animated contents," perhaps your confusion is why the paper mainly uses the anime example in Sec. 3. This is, in fact, a deliberate choice to improve the readability of the paper. As mentioned in the footnote on page 3, "The displayed video is the 1st from LVBench. We will consistently use it for method demonstrations for readers’ convenience." We thought that such consistency would make it easier for the reader to understand. Please let us know if you have further suggestions on this part.
>
> ## 4. Robustness of Character Naming and Intention Analysis
>
> **Character Diversity.** We are one of the first video agents trying to use VLMs for human-like consistent naming, and we indeed notice that the reliability of consistent naming is limited by the capabilities of existing VLMs, *e.g.*, Gemini-Flash. Even so, we notice that:
>
> 1. The VLMs are good at identifying celebrities or famous characters consistently, *e.g.*, the player Salah is recognized throughout the video in the example of Fig. 7 (left).
> 2. For less well-known characters, VLMs primarily rely on facial features to conduct the matching. For example, we checked the example video from Fig. 7 (right), including the later video parts that are not presented in the paper due to space limits: the identification of Mikhail is consistent, despite his clothes being different in the later parts of the video.
>
> Although we have observed several successful cases, we acknowledge the inherent challenge of consistent naming. Specifically, we check the captions for the video shown in Fig. 2, where the main character has demonstrated multiple ages, clothing, and camera angles. Although our pipeline manages to merge some of them, it is hard for the VLM to realize that these are different stages of the same character. Therefore, we hope our study can initiate the investigation into long-term character consistency by illustrating its utility in video QA (Fig. 6), so that this problem can be better noticed and addressed in future works via better frameworks and VLMs.
>
> **Intention Analysis.** We assume the reviewer is asking about whether the intention analysis depends on dialogues. In fact, most of the benchmarks we evaluated, such as LVBench, do not use dialogue. Therefore, our MR. Video mainly uses the visual content and the logic of the video to analyze which segment is relevant to the questions in the intention analysis. From this perspective, our framework is robust to videos with minimal dialogue.

---

### Note · Authors · 2025-08-13

We thank the reviewers, AC, and SAC for the effort and insightful feedback! We are grateful that the rebuttal process has resulted in **all positive scores**, without any outstanding concerns from the reviewers. Specifically, Reviewers b2Lt and wmFm increased their scores after their concerns were addressed, and Reviewers FvH9 and Zpo3 maintained their positive scores without raising new concerns.

Our MR. Video proposes a **novel MapReduce principle** for video agents, which marks a significant conceptual shift from the iterative key frame selection in previous video agents to **comprehensive context understanding** and **system-wise inference-time scalability**. Our principle, along with novel video understanding designs, leads to **significant advantages over the previous video agents**, represented by a **19% accuracy improvement** over VCA on LVBench.

During the rebuttal, we have clarified two main contexts regarding the reviewers' concerns:

- **Gemini-2.0-Flash vs. GPT-4o**. We have confirmed that our improvement is method-driven and the comparison is fair, because:
  - Gemini-2.0-Flash used in MR. Video is generally considered a weaker model than GPT4o used by previous video agents, with a 1/250 cheaper price.
  - MR. Video shows consistent advantages when evaluating previous agents with Gemini-2.0-Flash.
  - MR. Video also shows improvement on open-source models, *e.g.*, Qwen2.5-VL.
- **Inference Scalability**. The MapReduce principle introduced by our MR. Video shows better inference scalability than the iterative frame selection of previous video agents, because:
  - It is **necessary to scale up the tokens for a comprehensive understanding of the contexts**. We justify this design with both the example in Fig. 1 and our significant performance gains.
  - From the perspective of the critical path in system design, MR. Video perceives video segments **in parallel**, leading to **more controllable critical paths**. This enables better scalability: **the duration of videos that could be processed in unit time can scale linearly with the number of GPUs**.

We will integrate these discussions in the revision to better demonstrate our insights. We are confident that introducing the MapReduce principle to long video understanding marks a paradigm shift towards **comprehensive context understanding** and enabling **scalability from a system-wide parallelism perspective**. Thank you again for your time and feedback!

---

### Decision · Program_Chairs · 2025-09-17

**Decision:**

Accept (poster)

**Comment:**

All reviewers found the adaptation of the map-reduce formulation to video understanding to be interesting, novel, and well-presented, while acknowledging the strong empirical results demonstrated across a carefully selected collection of appropriate datasets. Although the original reviews raised several concerns regarding the use of proprietary closed models, consistent character naming conventions, token count considerations, and evaluation fairness, the authors' rebuttals successfully addressed these issues, with all reviewers expressing satisfaction with the responses provided. The area chairs agree that this submission represents a valuable and compelling contribution to the field of long-form video understanding, offering both methodological innovation and solid experimental validation.